# Single-cell dissection of prognostic architecture and immunotherap response in *Helicobacter pylori* infection-associated gastric cancer

Xin Zhang[1,2†], Guangyu Zhang[3†], Shuli Sang[1†], Yang Fei[4†], Xiaopeng Cao[5†], Wenge Song[1], Feide Liu[4], Jinze Che[4], Haoxia Tao[1], Hongwei Wang[6], Lihua Zhang[6], Yiyan Guan[1], Shipeng Rong[4], Lijuan Pei[4], Sheng Yao[4*], Yanchun Wang[1*], Min Zhang[7*], Chunjie Liu[1*]

[1]State Key Laboratory of Pathogen and Biosecurity, Institute of Biotechnology, Academy of Military Medical Sciences, Beijing, China; [2]Department of Pharmacy, Medical Supplies Center, Chinese PLA General Hospital, Beijing, China; [3]Center for Immune Ageing and Rejuvenation, Department of Rheumatology and Immunology, The First Affiliated Hospital of Chongqing Medical University, Chongqing, China; [4]Department of General Surgery, First Medical Center, Chinese PLA General Hospital, Beijing, China; [5]Department of Gastroenterology and Hepatology, First Medical Center, Chinese PLA General Hospital, Beijing, China; [6]Department of Pathology, Fourth Medical Center, Chinese PLA General Hospital, Beijing, China; [7]Department of Nephrology, First Medical Center of Chinese PLA General Hospital, Beijing, China

**\*For correspondence:**
ys848388@163.com (SY);
springwyc@163.com (YW);
mzhangcool@163.com (MZ);
liucj@bmi.ac.cn (CL)

[†]These authors contributed equally to this work

## eLife Assessment

This study presents a **valuable** description of the cellular and transcriptional landscape of the tumor microenvironment in 27 gastric cancer (GC) patients based on their *H. pylori* status (HpGC, ex-HpGC, non-HpGC). The single-cell RNA sequencing dataset and computational analysis are **convincing** and provide a starting point that is of value for understanding *H. pylori*-associated GC cell type composition, cell transitions, and mechanisms of response to therapy. The section correlating immunotherapy outcomes with GC cell type compositions from bulk RNAseq would have been strengthened by further comparing *H. pylori* GC versus non-*H. pylori* GC.

**Abstract** Most of the human gastric cancer (GC) worldwide are ascribed to *Helicobacter pylori* infections, which have a detrimental effect on the immunotherapy's efficacy. Comprehensively dissecting the key cell players and molecular pathways associated with cancer immunotherapies is critical for developing novel therapeutic strategies against *H. pylori* infection-associated human GC. We performed a comprehensive single-cell transcriptome analysis of nine GC patients with current *H. pylori* infection (HpGC), three GC patients with previous *H. pylori* infection (ex-HpGC), six GC patients without *H. pylori* infection (non-HpGC), and six healthy controls (HC). We also investigated key cell players and molecular pathways associated with GC immunotherapy outcomes. We revealed the molecular heterogeneity of different cell components in GC, including epithelium, immune cells, and cancer-associated fibroblasts (CAFs) at the single-cell level. The malignant epithelium of HpGC exhibited high expression level of inflammatory and epithelial–mesenchymal transition signature, HpGC and ex-HpGC were enriched with VEGFA+ angiogenic tumor-associated macrophages (Angio-TAM) and IL11+ inflammatory CAF (iCAF), characterized by high expression levels of

NECTIN2 and VEGFA/B. Additionally, we found significant correlations between the abundance of iCAF with Angio-TAM and TIGIT+ suppressive T cells, and iCAF interacted with Angio-TAM through the VEGF and ANGPTL angiogenic pathways. We also developed an immune signature and angiogenic signature and demonstrated that the iCAF abundance and angiogenic signature could predict poor immunotherapy outcomes in GC. We revealed the transcriptome characteristics and heterogeneity of various cellular constituents of HpGC patients and demonstrated that a synergistic combination of immunotherapy and anti-angiogenic targeted therapy may be an effective therapeutic modality for HpGC patients.

## Introduction

Gastric cancer (GC) is the fifth most diagnosed tumor and the third most common cause of cancer-causing death. GC is characterized by heterogeneous cellular characteristics and a tumor microenvironment (TME) comprising complex components, posing challenges to personalized therapy (*Oya et al., 2020*). The development of GC can be attributed to the long-term combined effects of lifestyle, environmental factors, genetic factors, and pathogenic infections. *Helicobacter pylori* infection (*Uemura et al., 2001*; *Hsu et al., 2007*) is an important carcinogenic factor and responsible for approximately 90% of noncardia GC worldwide and approximately 5% of the total burden from all cancers globally (*Plummer et al., 2015*). It also plays a critical role in TME regulation and is linked to response to immunotherapies (*Oster et al., 2022*; *Hatakeyama, 2006*). According to the histological examination, serology test, rapid urease test, and *H. pylori* DNA validation, *H. pylori* infection-associated GC cases can be classified into current *H. pylori* infection and past *H. pylori* infection, and the *H. pylori* infection status correlated with different molecular characteristics (*Kwak et al., 2014*; *Son et al., 2022*).

Immune checkpoint inhibitor (ICI)-targeted therapy is an effective anticancer treatment for a wide range of human malignancies (*Thompson, 2018*; *Carlino et al., 2021*; *Bagchi et al., 2021*; *Doroshow et al., 2021*). However, their efficacy in GC remains controversial (*Shi et al., 2022*). Previous clinical trial studies such as JAVELIN Gastric 300 (*Bang et al., 2018*), KEYNOTE-061 (*Shitara et al., 2018*), and KEYNOTE-062 (*Shitara et al., 2020*) have shown that PD-1/PD-L1 inhibitors do not have significant advantages over conventional chemotherapy in the treatment of GC. In contrast, the Check-Mate-649 study has revealed that the continued use of nivolumab plus chemotherapy as a standard first-line treatment achieved great success in advanced gastroesophageal adenocarcinoma (*Shitara et al., 2022*). A subgroup analysis has revealed that the Asian population benefits more from immunotherapy than the global population, and among the Asian population, the Chinese population benefits the most from immunotherapy, which could be ascribed to the molecular features leading to GC in Chinese patients, such as *H. pylori* infection (*Zhang et al., 2022*; *Peng et al., 2020*). However, recent studies have demonstrated that *H. pylori* infection hindered the efficacy of immune checkpoint-targeting PD-1/PD-L1 inhibitors or vaccine-based immunotherapies through immune evasion, T cell suppression, and fluctuation in the intestinal flora, compromising the benefits of cancer immunotherapies (*Oster et al., 2022*; *Shi et al., 2022*). The cellular and molecular heterogeneity of *H. pylori* infection-associated GC prevent effective immunotherapies; thus, revealing the discrepancy in the molecular characteristics of various cellular components among GC patients with different *H. pylori* infection status is required for accurate diagnosis and treatment.

The continued availability of single-cell RNA sequencing (scRNA-seq) has allowed systematic profiling of dynamic transcriptional characteristics of thousands of cells simultaneously at the single-cell level with high throughput and low cost, enabling unbiased deciphering of the molecular heterogeneity of various components in biological tissues (*Gohil et al., 2021*; *Wu and Swarbrick, 2021*). A previous study has applied scRNA-seq technology in various types of gastric mucosa biopsies with different lesions to reveal dynamic transcriptional changes in the cascade of GC and to construct a transcriptional regulatory network in the gastric epithelium (*Zhang et al., 2019*). We have also previously constructed a systematic transcriptomic landscape of gastric adenocarcinoma and profiled the developmental trajectory of GC (*Zhang et al., 2021*). Additionally, scRNA-seq has been used to dissect TME heterogeneity (*Wang et al., 2021*) and molecular features of various components in the GC TME, including immune cell diversity, which facilitates the understanding of tumor biology and sheds light on novel targets for immunotherapy (*Fu et al., 2020*; *Sathe et al., 2020*; *Kim et al.,*

*2022a*). However, little attention has been paid to the relationship between *H. pylori* infection, diversity of GC TME, and immunotherapy response at the single-cell level.

This study aimed to reveal the diversity of the *H. pylori* infection-associated GC TME and distinguish key cellular players, molecular pathways, and effector programs associated with response to immunotherapies in GC. We performed unbiased scRNA-seq on gastric tissues of healthy control (HC, n=6), GC without *H. pylori* infection (non-HpGC, n=6), GC with current *H. pylori* infection (HpGC, n=9), and GC with previous *H. pylori* infection (ex-HpGC, n=6) to present the first cellular atlas, including a total of 83,637 high-quality cells for deciphering *H. pylori* infection-associated transcriptomic architecture. We also profiled the intercellular crosstalk among suppressive T cells, angiogenic tumor-associated macrophages (Angio-TAMs), and inflammatory cancer-associated fibroblasts (iCAFs), which correlated with prognosis and response to immunotherapy. In particular, iCAFs that were linked to Angio-TAM and suppressive T cells exhibited upregulation of NECTIN2 and VEGFA/B (vascular endothelial growth factor A/B), which is indicative of poor immunotherapy efficacy. Moreover, intercellular crosstalk analysis revealed that iCAFs may promote tumor angiogenesis and immune suppression in *H. pylori* infection-associated GC, by intercellular interaction with Angio-TAM through the VEGFA/B-VEGFR1 pathway, and TIGIT[+] suppressive T cells through the NECTIN2-TIGIT pathway, respectively. Our study indicates that the combination of immunotherapy and vascular-targeted therapies could be a potentially efficient approach for the treatment of *H. pylori* infection-associated GC patients.

## Results

### Single-cell transcriptomic architecture and molecular features of *H. pylori* infection-associated GC

In this study, we first classified the collected GC tissues into non-HpGC, HpGC, and ex-HpGC to accurately evaluate the *H. pylori* infection status of GC patients, which was defined by a combination of serology examinations along with the *H. pylori* DNA assay and histological examination (HE) of each gastric tissue ('Materials and methods', *Table 1*). The gastric tissues selected were collected from six healthy donors (HCs) and six non-HpGC, nine HpGC, and six ex-HpGC patients, who were recently diagnosed and did not receive chemotherapy and radiotherapy before surgery (*Figure 1A*). Viable cells with high quality were collected from each gastric tissue for further scRNA-seq using fluorescence-activated cell sorting (FACS). A total of 83,637 cells were retained for subsequent analysis after rigorous quality control, which yielded an average of 1358 genes and 3465 transcripts in each cell (*Figure 1—figure supplement 1 A–D*). We first employed the inferCNV to evaluate copy number variation (CNV) of autosomal genes among all cells to distinguish malignant and non-malignant cells (*Figure 1—figure supplement 1E*), We then identified and annotated nine main cell types according to the expression of canonical gene markers and CNV distribution, which were composed of the non-malignant epithelium (marked with *MUC5AC*, *GKN1*, and *TFF1/2*), malignant epithelium (*KRT17*, *S100A9* and *REG4*), endothelium (marked with *PECAM1*, *VWF*, and *PLVAP*), fibroblasts (marked with *DCN*, *LUM,* and *COL1A1*), plasma cells (marked with *JCHAIN* and *CD49A*), monocyte/macrophage (marked with *C1QA/B/C* and *IL1B*), mast cells (marked with *CPA3* and *TPSAB1*), B cells (marked with *CD79A* and *MS4A1*), and T cells (marked with *CD2*, *CD3D*, and *CD3E*) (*Figure 1B–E*, *Figure 1—figure supplement 1F*, and *Supplementary file 1*). To investigate the influence of *H. pylori* infection on the heterogeneity of gastric TME, we analyzed the dynamic cellular component changes of gastric TME in non-HpGC, ex-HpGC and HpGC compared with that of HC. The results revealed a significant increase in the proportion of T cells in GC, including non-HpGC, ex-HpGC, and HpGC, in contrast with HC (*Figure 1F and G*), which aligned with the consensus that tumorigenesis is accompanied by inflammation. In addition, the percentage of non-malignant epithelium significantly decreased in GC compared to normal HC. Whereas there was a significant increase in the proportion of malignant epithelium in GC compared with that of HC, which is closely related to the process of carcinogenesis. However, there is no significant difference in the proportion of non-malignant epithelium and malignant epithelium among the non-HpGC, ex-HpGC, and HpGC (*Figure 1G*). To further elucidate the correlation between *H. pylori* infection and the transformation of non-malignant epithelium into malignant epithelium during gastric carcinogenesis, we performed a comparative analysis of upregulated genes characteristics between non-malignant epithelial cells and malignant epithelial cells. The

**Table 1.** Patient characteristics of each sample in gastric cancer (GC) scRNA-seq.

| Sample | Stage | Age | Sex | Histopathological diagnosis | Site of origin | Lauren's classification | H. pylori serum antibody | H. pylori DNA | H. pylori cagA | H. pylori on H&E slide |
|---|---|---|---|---|---|---|---|---|---|---|
| exHpGC1 | III | 62 | F | Moderately differentiated adenocarcinoma | Gastric antrum and corpus | Intestinal | + | - | - | - |
| exHpGC2 | III | 61 | M | Moderately differentiated adenocarcinoma | Gastric antrum | Mixed | + | - | - | - |
| exHpGC3 | I | 75 | M | Poorly differentiated adenocarcinoma | Gastric antrum | Diffuse | + | - | - | - |
| exHpGC4 | II | 67 | F | Signet ring cell carcinoma | Gastric body | Diffuse | + | - | - | - |
| exHpGC5 | III | 64 | M | Moderately poorly differentiated adenocarcinoma | Cardia | Intestinal | + | - | - | - |
| exHpGC6 | I | 65 | M | Moderately differentiated adenocarcinoma | Gastric antrum | Intestinal | + | - | - | - |
| HpGC1 | II | 73 | M | Moderately poorly differentiated adenocarcinoma | Stomach angle | Intestinal | + | + | - | + |
| HpGC2 | IV | 55 | M | Poorly differentiated adenocarcinoma | Gastric antrum and corpus | Diffuse | + | + | - | + |
| HpGC3 | I | 49 | F | Moderately-well differentiated adenocarcinoma | Gastric antrum and gastric angle | Intestinal | + | + | - | + |
| HpGC4 | I | 47 | M | Poorly differentiated adenocarcinoma, mostly signet ring cell carcinoma | Gastric antrum | Diffuse | + | + | - | + |
| HpGC5 | II | 67 | F | Signet ring cell carcinoma | Gastric antrum | Diffuse | + | + | - | + |
| HpGC6 | I | 58 | M | Moderately poorly differentiated adenocarcinoma | Greater curvature | Mixed | + | + | + | + |
| HpGC7 | III | 67 | F | Moderately poorly differentiated adenocarcinoma | Angular incisure | Intestinal | + | + | N/A | + |
| HpGC8 | III | 74 | M | Moderately poorly differentiated adenocarcinoma | Greater curvature | Intestinal | + | + | N/A | + |
| HpGC9 | II | 62 | M | Poorly differentiated adenocarcinoma, partial signet ring cell carcinoma | Angular incisure | Mixed | + | + | N/A | + |
| nonHpGC1 | I | 54 | F | Poorly differentiated adenocarcinoma | Cardia | Diffuse | - | - | N/A | - |
| nonHpGC2 | III | 65 | M | Moderately differentiated adenocarcinoma | Cardia | Intestinal | - | - | N/A | - |

*Table 1 continued*

| Sample | Stage | Age | Sex | Histopathological diagnosis | Site of origin | Lauren's classification | H. pylori serum antibody | H. pylori DNA | H. pylori cagA | H. pylori on H&E slide |
|---|---|---|---|---|---|---|---|---|---|---|
| nonHpGC3 | III | 56 | M | Moderately differentiated adenocarcinoma | Cardia | Intestinal | - | - | N/A | - |
| nonHpGC4 | III | 72 | M | Moderately poorly differentiated adenocarcinoma | Angular incisure | Intestinal | - | - | N/A | - |
| nonHpGC5 | I | 56 | M | Poorly differentiated adenocarcinoma | Antrum | Mixed | - | - | N/A | - |
| nonHpGC6 | II | 63 | M | Poorly differentiated adenocarcinoma | Angular incisure | Mixed | - | - | N/A | - |
| HC1 | N/A | 38 | M | Normal | Greater curvature | N/A | - | - | N/A | - |
| HC2 | N/A | 43 | F | Normal | Greater curvature | N/A | - | - | N/A | - |
| HC3 | N/A | 62 | F | Chronic gastritis | Gastric body | N/A | - | - | - | - |
| HC4 | N/A | 61 | F | Normal | Gastric antrum | N/A | - | - | - | - |
| HC5 | N/A | 25 | M | Normal | Gastric antrum | N/A | - | - | - | - |
| HC6 | N/A | 32 | F | Normal | Gastric antrum | N/A | - | - | - | - |

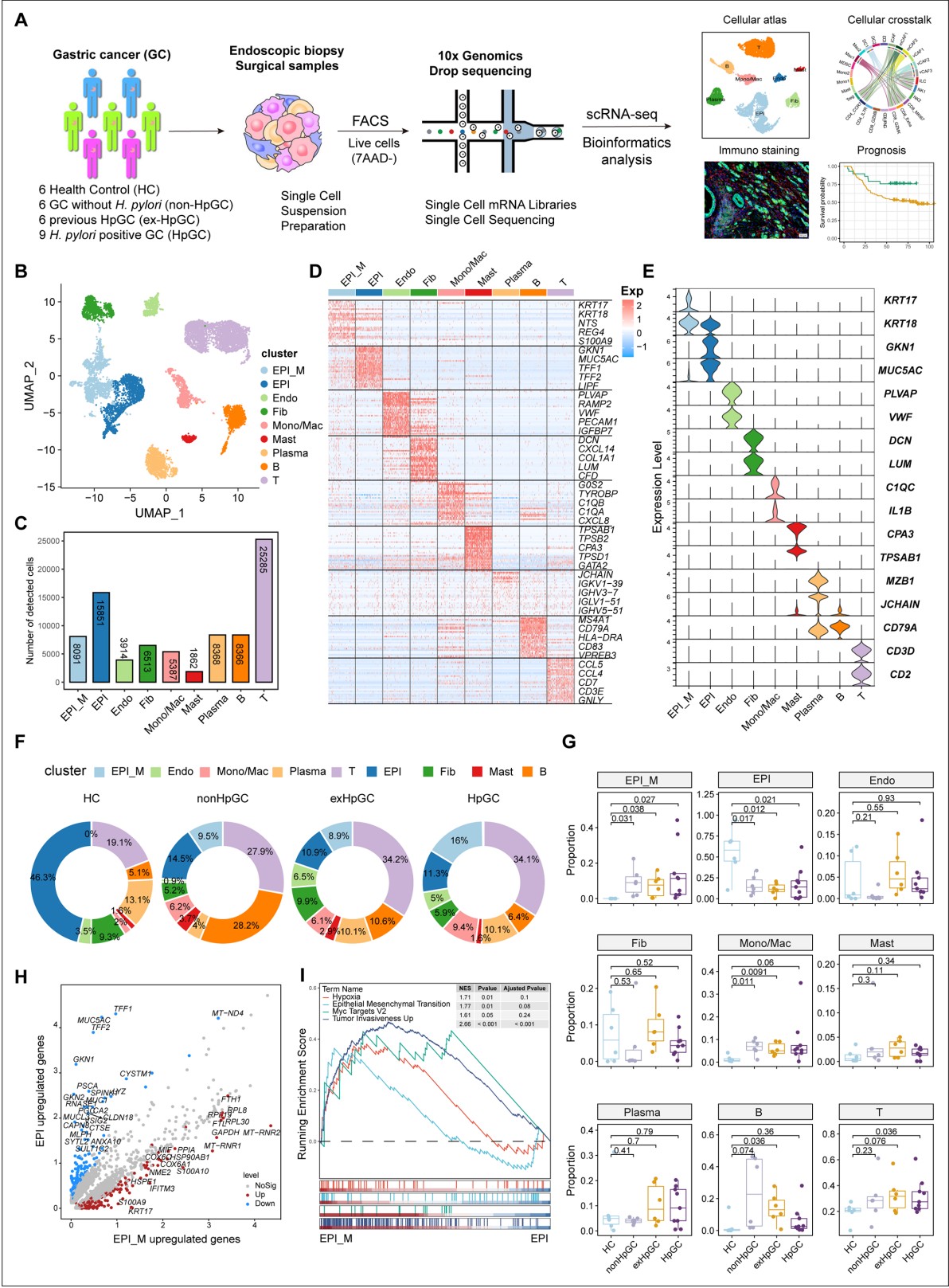

**Figure 1.** Global analysis of the tumor microenvironment and malignant features of *H. pylori* infection-associated gastric cancer (GC). (**A**) A general workflow of GC sample preparation and processing of single-cell suspensions for scRNA-seq analysis. In total, 27 gastric samples, including gastric tissues of healthy control (HC, n=6), GC without *H. pylori* infection (non-HpGC, n=6), GC with previous *H. pylori* infection (ex-HpGC, n=6), and GC with current *H. pylori* infection (HpGC, n=9), were collected to perform scRNA-seq. (**B**) Uniform Manifold Approximation and Projection (UMAP) plot

*Figure 1 continued on next page*

*Figure 1 continued*

for unbiased clustering and cell type annotation of 86,637 high-quality cells. EPI_M: malignant epithelium; EPI: non-malignant epithelium; Endo: endothelium; Fib: fibroblast; Mono/Mac: monocyte/macrophage; Plasma: plasma cells; Mast: mast cells; B: B cells; T: T cells. (**C**) The absolute quantities of nine different cell types. (**D**) Heatmap showing the top eight differentially expressed genes (DEGs) of nine main cell types. (**E**) Violin plots showing the expression of signature genes of nine cell types. (**F**) Pie plot revealing the proportions of nine cell types in HC, non-HpGC, ex-HpGC and HpGC. (**G**) Box plot showing statistical analysis of proportion of nine cell types in HC, non-HpGC, ex-HpGC, and HpGC. The p-value of Student's *t*-test is shown. (**H**) Volcano plot displaying the differentially upregulated genes in EPI and EPI_M. (**I**) Gene set enrichment analysis (GSEA) showing the pathway activity (scored per cell by GSEA) in malignant epithelium (EPI_M) and non-malignant epithelium (EPI).

The online version of this article includes the following figure supplement(s) for figure 1:

**Figure supplement 1.** Data quality control and filtering criteria of gastric cancer (GC) scRNA-seq.

result revealed that upregulated genes in the non-malignant epithelium included *MUC5AC*, *GKN1*, and *TFF2*, which participate in the secretion of stomach mucus and digestive enzymes, whereas upregulated genes in the malignant epithelium included *TM4SF1*, *IFITM3*, *IGFBP7*, and *KRT17* (*Figure 1H*, *Figure 1—figure supplement 1F and G*). Gene set enrichment analysis (GSEA) showed that the differentially upregulated genes in the malignant epithelium compared to non-malignant epithelium were enriched in pathways associated with tumor development and progression, such as hypoxia, tumor invasiveness, MYC targets, and epithelial–mesenchymal transition (EMT) (*Figure 1I*). In brief, our results provided a comprehensive landscape of the cellular and transcriptomic diversity within the cell components in gastric TME-associated *H. pylori* infection.

## Cellular characterization of the malignant epithelium of *H. pylori* infection-associated GC

To further explore the molecular features of *H. pylori* infection-associated GC and the heterogeneity of the malignant epithelium, we performed a subclustering analysis of malignant cells and generated six distinct malignant epithelium subpopulations (M1–M6), in which clusters M1 and M2 were mainly derived from non-HpGC, whereas clusters M5 and M6 were mainly originated from *H. pylori* infection-associated GC samples including HpGC and ex-HpGC (*Figure 2A and B*, *Figure 2—figure supplement 1A and B*). The expression distribution of these clusters was validated in HpGC by deconvolution analysis using bulk RNA sequencing datasets (Wilcoxon test; *Figure 2—figure supplement 1C*; *Cristescu et al., 2015*). We found that malignant epithelium of M1/M2/M3/M4 exhibited high expression levels of epithelium differentiation-related genes, such as *PHGR1* and *KRT20* (*Figure 2C*), whereas M5/M6 or *H. pylori* infection-associated GC samples exhibited high expression levels of inflammation-related genes, such as *CXCL1/2/3/8*, *S100A9*, *S100P*, and EMT signature, such as *SPARC* and *VIM* (*Figure 2C and D*). In addition, among the *H. pylori* infection-associated GC samples, the expression of inflammation-related genes and IgG family members' genes was much higher in HpGC (*Figure 2D*, *Figure 2—figure supplement 1*, and *Supplementary file 2*). Interestingly, GSEA of the upregulated genes in the HpGC malignant epithelium was mainly associated with EMT, epithelial cell proliferation, and inflammatory response pathways (*Figure 2E*). We also found that the malignant epithelium exhibited a high degree of differentiation inter- and intra-heterogeneity (*Figure 2F and G*), the differentiation score of ex-HpGC2 and ex-HpGC5 presented two distinct peaks, one of which is comparable to the peak observed in HpGC, while the another of which resembled the peak observed in non-HpGC sample (*Figure 2F*). The finding suggested that a history of *H. pylori* infection in GC can still contribute to the differentiation heterogeneity of malignant epithelium. Furthermore, malignant epithelium from ex-HpGC displayed low differentiation scores, which were even lower in malignant epithelium of HpGC (p<0.05, Wilcoxon test; *Figure 2H*). This further substantiates the correlation between the inflammatory and differentiation ability with the severity of *H. pylori* infection. Furthermore, the abundance of malignant epithelium clusters could predict different prognosis outcomes (*Figure 2—figure supplement 1E*), and the high differentiation score is highly associated with a better prognosis of GC and vice versa (*Figure 2I*).

Collectively, the findings highlight the transcriptomic heterogeneity of malignant epithelium with different *H. pylori* infection status and the potential utility of the molecular features within the malignant epithelium to distinguish HpGC from non-HpGC, indicative of a high degree of tumor heterogeneity and the influence of the *H. pylori* infection on gastric carcinogenesis and prognosis.

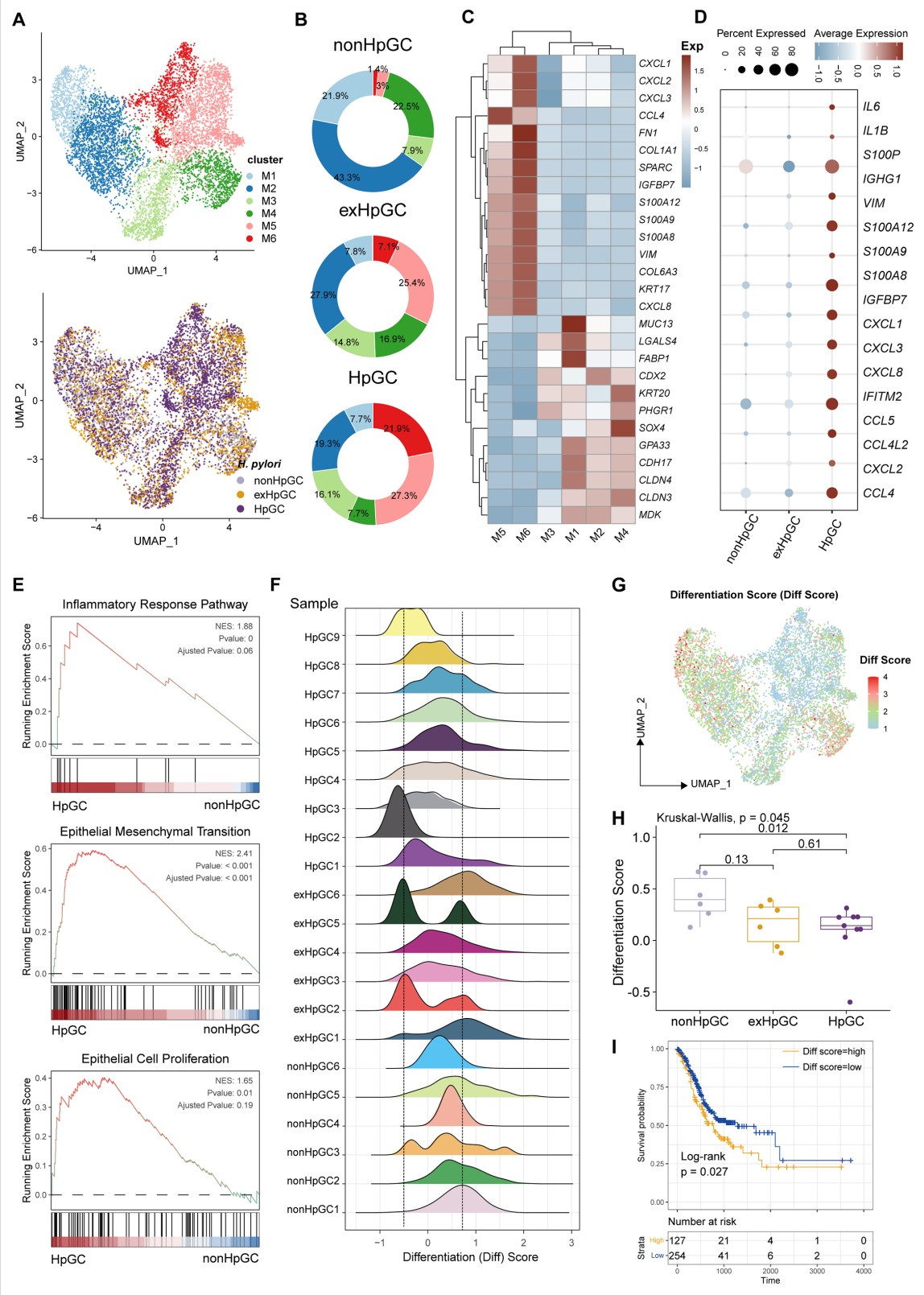

**Figure 2.** Characterization of the malignant epithelium within gastric cancer (GC) with different *H. pylori* infection status. (**A**) Uniform Manifold Approximation and Projection (UMAP) plot showing six subpopulations of malignant epithelium, colored by different cell types (upper) and different *H. pylori* infection status (lower). (**B**) Pie plot showing the proportion of six subsets of malignant epithelium in non-HpGC, ex-HpGC, and HpGC. (**C**) Heatmap displaying the differentially expressed genes (DEGs) among the six subsets of malignant epithelium. (**D**) Bubble plot showing the

*Figure 2 continued on next page*

*Figure 2 continued*

difference in representative molecular among the non-HpGC, ex-HpGC, and HpGC. (**E**) Gene set enrichment analysis (GSEA) showing the pathway activity (scored per cell by GSEA) in malignant epithelium of HpGC compared to that of non-HpGC. NES, normalized enrichment score. (**F**) The ridge plot showing the differentiation score (Diff Score) of malignant epithelium within each sample. (**G**) Heatmap showing the Diff Score of six subpopulations of malignant epithelium. (**H**) Box plot showing differentiation score among non-HpGC, ex-HpGC, and HpGC, p-values were assessed by Wilcoxon test, two-way ANOVA test is used for comparison of multiple groups. (**I**) Kaplan–Meier survival analysis of The Cancer Genome Atlas (TCGA) stomach adenocarcinoma (STAD) patients stratified by tumor sample differentiation scores, which was used to group samples into high and low groups based on 33rd and 67th percentile. The p-value of two-sided log-rank test is shown.

The online version of this article includes the following figure supplement(s) for figure 2:

**Figure supplement 1.** Malignant epithelium characteristic in gastric cancer (GC) with different *H. pylori* infection status.

## Cellular characterization of the non-malignant epithelium of *H. pylori* infection-associated GC

To determine the cellular phenotype and composition of the gastric mucosa during the progression of *H. pylori* infection, we performed a transcriptomic analysis of the non-malignant epithelium, and these cells were partitioned into nine cell types (*Figure 3A*), including chief cells (marked by *PGA3* and *PGA4*), neck cells (marked by *MUC6*), spasmolytic polypeptide-expressing metaplasia (SPEM, marked by *MUC6* and *TFF2*), intestinal metaplasia (IM, marked by *MUC2*), enterocytes (marked by *APOC3* and *FABP1*), endocrine cells (marked by *CHGA*), pit mucous cell (PMC, marked by *GKN1/2* and *MUC5AC*), pre-PMC (marked by medium expression level of *MUC5AC* and *GKN2*), and parietal cells (marked by *ATP4A*) (*Figure 3B and C*, *Supplementary file 3*). We compared the composition of cells in normal mucosae with that of GC mucosae and observed that the neck cell (p=0.034, Student's *t*-test), chief cell (p=0.014, Student's *t*-test), and parietal cells (p=0.0087, Student's *t*-test) decreased dramatically in HpGC compared to HC, whereas the HpGC exhibited increase partition of metaplasia lineage SPEM (p=0.017, Student's *t*-test) and intestinal-specific cell types, such as IM (p=0.036, Student's *t*-test) and enterocytes (p=0.036, Student's *t*-test). Furthermore, HpGC samples had even higher proportion of metaplasia lineage SPEM and intestinal-specific cell types and a lower proportion of neck cells and parietal cells than that of ex-HpGC and non-HpGC (*Figure 3C*). To conform the validity of our findings, we performed deconvolution analysis with independent microarray data (GSE2669), including 124 normal mucosa control (NC), chronic gastritis (CG), intestinal metaplasia (IM), GC associated with *H. pylori* infection samples to decipher the changes in the proportion of cell components with different stages of gastric tissue. Consistent with our scRNA-seq findings, the proportion of SPEM, IM, and enterocytes in GC increased gradually during the gastric carcinogenesis process, with even higher proportion in GC compared with that of CG and IM. In contrast, the proportion of chief cells, neck cells, and parietal cells decreased significantly in GC, with the lowest proportion in GC compared with that of CG and IM (p<0.05, Student's *t*-test; *Figure 3D and E*). These findings further support the validity of our results and indicate that chronic atrophic gastritis induced by *H. pylori* infection is the main cause of premalignant lesions.

To explore the origin of the premalignant cellular lineage and possible transition mechanism, we performed a single-cell analysis to reveal the developmental trajectories of various cell types in the non-malignant epithelium of GC (*Figure 3F*). The results showed that chief cells would differentiate into necks, and then the developmental path would divide these cells into two branches, one of which would develop into SPEM, whereas the other represents SPEM that could further develop into IM and enterocytes through transition states under a persistent inflammatory stimulus (*Figure 3G*). In addition, the pattern of pseudotemporal dynamic expression of specific representative genes and transcription factors (TFs) also supports the transition of chief cells to SPEM and IM metaplasia status and finally enterocytes (*Figure 3H and I*).

In brief, our results reveal the transition of GC non-malignant epithelium from chief cells to neck cells, SPEM and IM, and finally to enterocytes, under chronic inflammation at the single-cell level, which may deepen our understanding of the role of *H. pylori* infection in GC.

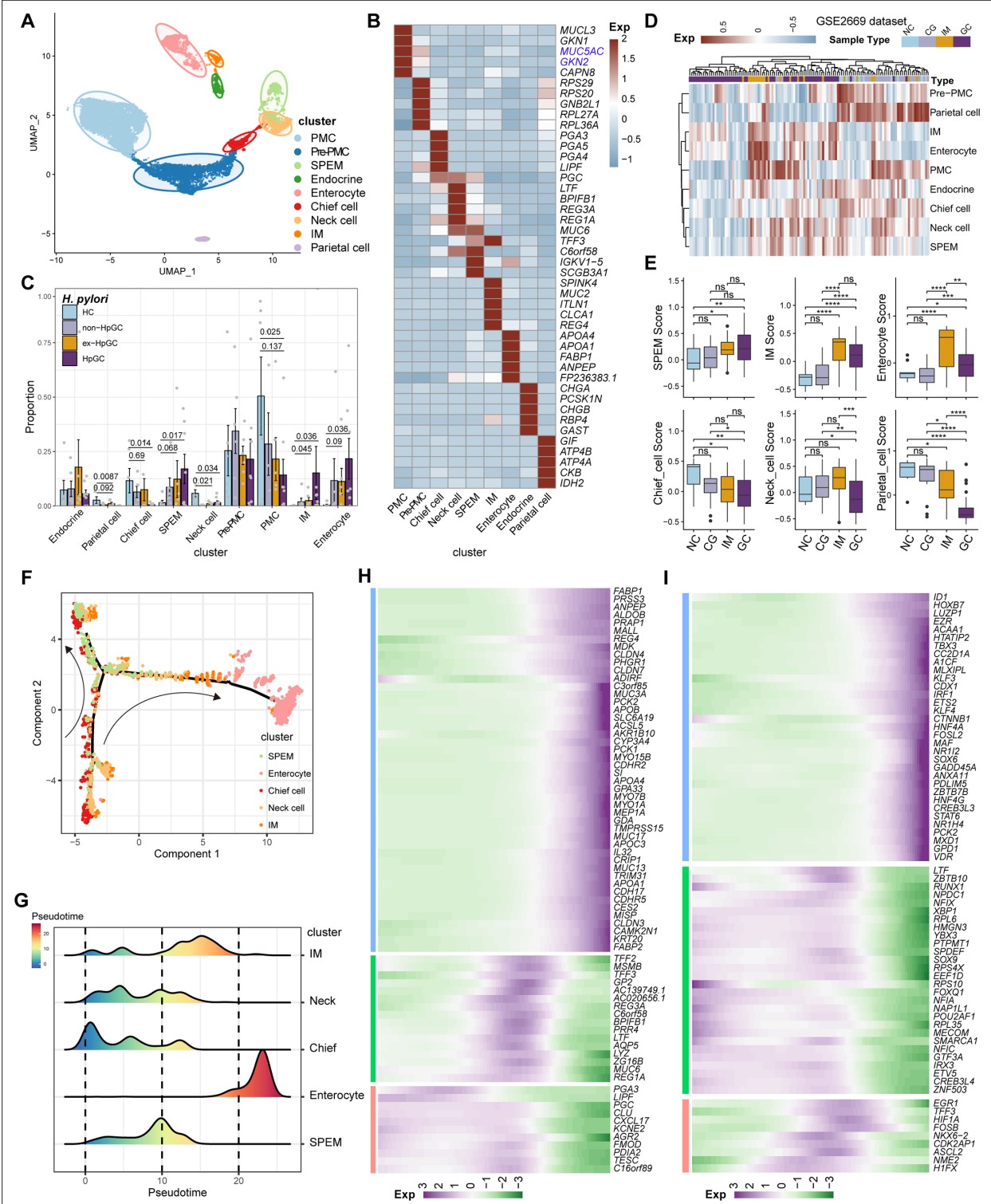

**Figure 3.** Characterization of the non-malignant epithelium within gastric cancer (GC) with different *H. pylori* infection status. (**A**) Unbiased clustering of non-malignant epithelium generated nine subtypes. (**B**) Heatmap showing the molecular feature of non-malignant epithelium according to the top five differentially expressed genes (DEGs). (**C**) Box plot showing the dynamic proportion of different cell types in non-malignant epithelium with different *H. pylori* infection status. (**D**) Heatmap showing the relative abundance (estimated by gene set variation analysis [GSVA]) of non-malignant epithelium subtypes in normal control (NC), chronic gastritis (CG), intestinal metaplasia (IM), GC samples (GSE2669). (**E**) Boxplot showing the relative abundance (estimated by GSVA) of non-malignant epithelium subtypes SPEM, IM, enterocytes, chief cells, neck cells, and parietal cells within NC, CG, IM, and GC samples (GSE2669). The p-value of Student's *t*-test is shown. (**F**) The trajectory analysis shows potential differentiation and transition trajectories in non-

*Figure 3 continued on next page*

*Figure 3 continued*

malignant epithelium clusters. (**G**) The ridge plot showing the pseudotime of non-malignant epithelium revealed the gastric pre-lesion process. (**H, I**) Heatmap showing scaled expression of dynamic genes (**G**) and TFs (**H**) along cell pseudotime.

## Cellular characterization of T lymphoid cells of *H. pylori* infection-associated GC

Tumor-infiltrating lymphocytes (TILs) are highly heterogeneous lymphocyte subsets. The activation of the different subsets of effector T cells influences the clinical outcome of *H. pylori* infection. Subclustering of T lymphoid cells and natural killer (NK) cells generated 11 main cell clusters in HC, non-HpGC, ex-HpGC, and HpGC (*Figure 4A*), including innate lymphoid cell (ILC, marked by *KIT*, *LST1*, and *ZBTB16*), NK1 (marked by *TYROBP*, *GZMA*, and *CEBPD*), NK2 (marked by *NKG7*, *CCL3*, *GZMH*, and *EOMES*), Treg (marked by *FOXP3*, *BATF*, and *TIGIT*), proliferative CD8_MKI67 (marked by *MKI67* and *CXCL13*), exhausted CD8_CXCL13 (marked by *CXCL13* and *TIGIT*), CD8_GZMK (marked by *GZMK*, *CCL4*, and *YBX3*), effective CD8_IFNG (marked by *IFNG*, *ATF3*), cytotoxic CD8_GZMB (marked by *GZMB*, *KLRC1*, and *MYBL1*), naïve CD4_IL7R (marked by *CCR7*, *SELL*, and *CD40LG*), and CD4_CCR7 (marked by *CCR7*, *SELL*, and *CD40LG*) (*Figure 4B and C*, *Supplementary file 4*). The organization of the T cell compartment showed great differences in GC with different *H. pylori* infection status (*Figure 4D*). The exhausted CD8_CXCL13, CD8_MKI67, and Tregs were prominently enriched in ex-HpGC and HpGC samples compared to those in HC ($p<0.05$, Student's *t*-test; *Figure 4E*), with a higher percentage in ex-HpGC and HpGC than in non-HpGC. The cytotoxic CD8_GZMB level was significantly decreased in ex-HpGC and HpGC ($p<0.05$, Student's *t*-test; *Figure 4E*), which coincides with previous research indicating immunosuppression in the TME. The deconvolution analysis using The Cancer Genome Atlas (TCGA) GC bulk dataset (*Bass and Thorsson, 2014*) indicated that the abundance of Treg and CD8_CXCL13 were elevated in HpGC than non-HpGC ($p=0.018$, and $= 0.12$, respectively, Wilcoxon test; *Figure 4F*). Next, we calculated the cytotoxic and inhibitory scores across various T lymphoid cell types, revealing that two subtypes of NK cells have the highest cytotoxic score, indicative of apoptosis, and Tregs have the highest inhibitory score, implying immunosuppression (*Figure 4G*). Overall survival analysis demonstrated that a high proportion of Tregs and CD8_CXCL13 represent a worse prognosis ($p=0.022$ and $=0.22$, respectively, log-rank test; *Figure 4H*). TIGIT, an important immune checkpoint with ligands including PVR and NECTIN2, is highly expressed in various immune cell types, particularly in Tregs within the TME, and is emerging as an immunotherapy target (*Chauvin and Zarour, 2020*; *Harjunpää and Guillerey, 2020*; *Ge et al., 2021*). Interestingly, we also found that the suppressive T cells, including Treg and CD8_CXCL13, interacted closely with the malignant epithelium through TIGIT-PVR/NECTIN2 ligand–receptor pairs, with even stronger interactions between suppressive T cells and malignant epithelium M4 and M6 (*Figure 4I*). Interestingly, NECTIN2 and PVR were more enriched in ex-HpGC and HpGC, compared with non-HpGC (*Figure 4J*). Immunostaining of HpGC also revealed that suppressive TIGIT[+] T cells were located close to the pan-CK[+] NECTIN2[+] malignant epithelium (*Figure 4K*, *Figure 4—figure supplement 1*), which supported the transcriptional findings that suppressive T cells could interact with the malignant epithelium through TIGIT–NECTIN2/PVR pairs. Collectively, we found that HpGC and ex-HpGC exhibits high expression levels of NECTIN2, PVR, and its target TIGIT, which may participate in immune escape in GC, indicated that blockade of TIGIT-NECTIN or TIGIT-PVR axes may be a promising immunotherapy modality for *H. pylori* infection-associated GC.

## Cellular characterization of myeloid cells of *H. pylori* infection-associated GC

Myeloid cells are also an important component of tumor-infiltrating immune cells, which play an important role in the regulation of tumor inflammatory responses and angiogenesis. Subclustering analysis of various myeloid cells in GC and normal gastric tissues (*Figure 5A*) revealed mast cells (marked by *CPA3*), neutrophils (marked by *CXCR2*, *IL1R2*), monocytes (marked by *S100A9*, *EREG*, and *VCAN*), FOLR2_TAM (marked by *LYVE1* and *FOLR2*), TREM2_TAM (marked by *TREM2*), C1QC_TAM (marked by *C1QC*), VEGFA[+]SPP1[+] Angio-TAM (marked by *VEGFA* and *SPP1*), and three subsets of dendritic cells (DCs, marked by *CD1C*, *LAMP3*, *IDO1*, and *IRF8*) (*Figure 5B and C*, *Figure 5—figure supplement 1A*). The proportion of TREM2_TAMs was significantly increased in HpGC and ex-HpGC,

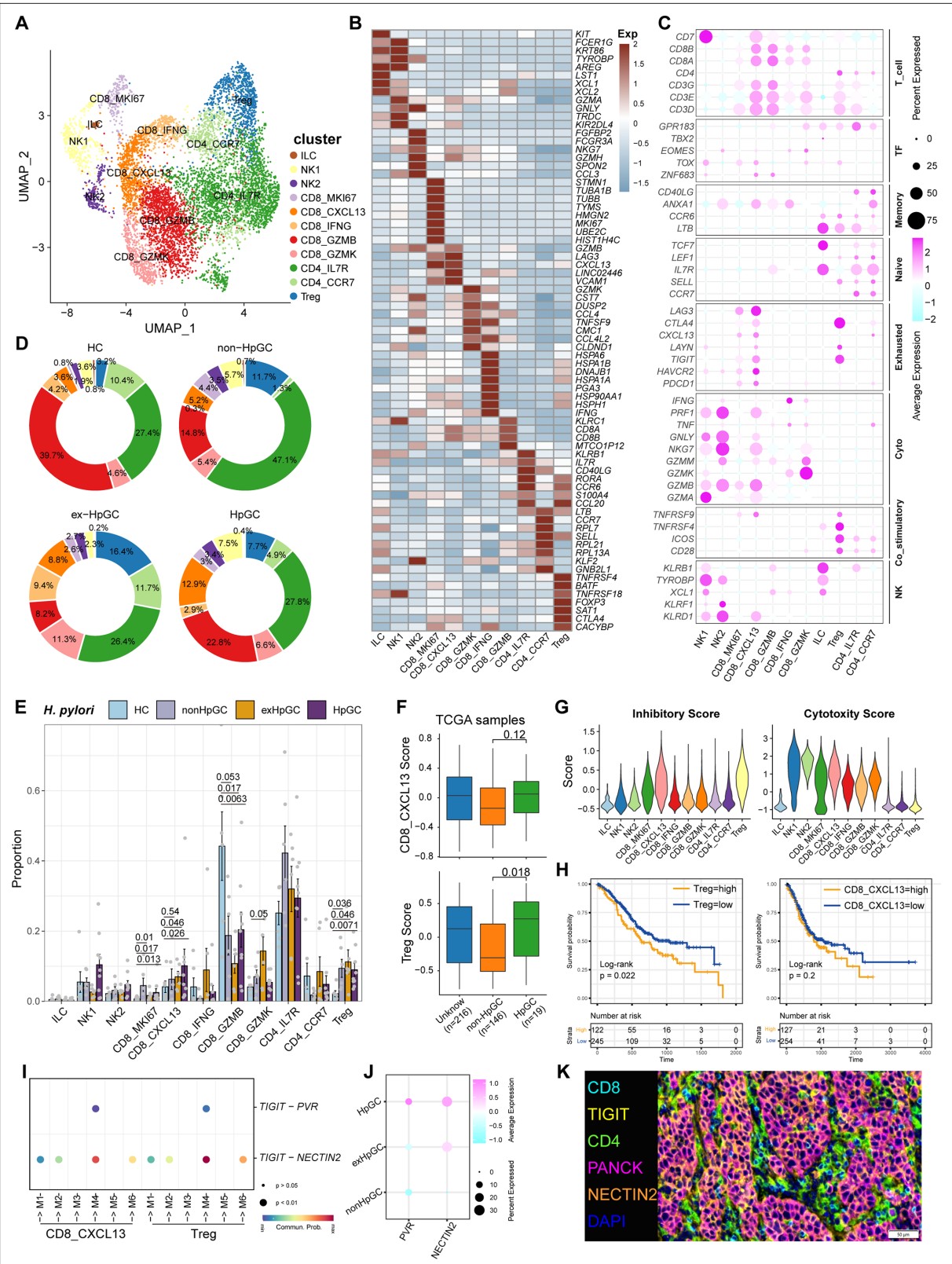

**Figure 4.** Characterization of tumor-infiltrating T and natural killer (NK) cells in *H. pylori* infection-associated gastric cancer (GC). (**A**) Unbiased clustering of T and NK cells generated 11 clusters. (**B, C**) Molecular features annotations according to the top five differentially expressed genes (DEGs) (**B**), and representative genes (**C**). (**D**) Pie plot showing the T/NK cell subtype abundance distribution in HC, non-HpGC, ex-HpGC, and HpGC. (**E**) The percentage contribution of T/NK cell subtype in HC, non-HpGC, ex-HpGC, and HpGC samples. p-Values were assessed by Student's *t*-test. (**F**) The

*Figure 4 continued on next page*

Figure 4 continued

deconvolution analysis showing the relative abundance of CD8_CXCL13 and Tregs in GC with different *H. pylori* infection status with The Cancer Genome Atlas (TCGA) stomach adenocarcinoma (STAD) dataset, p-values were assessed by Wilcoxon test. (**G**) The cytotoxic and inhibitory expression scores in T and NK clusters. (**H**) Kaplan–Meier survival analysis of TCGA STAD patients stratified by CD8_CXCL13 and Tregs relative abundance, which was used to group samples into high and low groups based on 33rd and 67th percentile. The p-value of two-sided log-rank test is shown. (**I**) Bubble plot showing intercellular interactions between suppressive T cells and malignant cells. (**J**) Dot plot showing the expression of NECTIN2 and PVR in malignant non-HpGC, ex-HpGC, and HpGC cells. (**K**) Immunostaining showing the ligand TIGIT expressed in suppressive T cells and the receptor NECTIN2 expressed on the malignant epithelium, respectively, in one HpGC sample.

The online version of this article includes the following figure supplement(s) for figure 4:

**Figure supplement 1.** Immunostaining of the ligand TIGIT and the receptor NECTIN 2 on suppressive T cells and the malignant epithelium, respectively, in HpGC, ex-HpGC, non-HpGC, and HC.

and Angio-TAMs were higher in HpGC tissues than in normal gastric tissues (p<0.05, Student's *t*-test; *Figure 5D*). TREM2_TAMs exhibited high expression levels of complement genes such as C1QA/ C1QB/C1QC, which are predictive factors of prognosis, whereas Angio-TAMs expressed chemokines such as CXCL3, CXCL5, CXCL8, and IL1RN, which are involved in the immune escape (*Lin et al., 2019*; *Figure 5E* and *Supplementary file 5*). Further signature enrichment analysis revealed that Angio-TAMs exhibited high expression levels of angiogenesis signature, whereas TREM2_TAMs exhibited a high level of M2 signature (*Figure 5F*). Immunostaining revealed that Angio-TAMs were mainly located in the tumor vasculature, whereas TREM2_TAMs were mainly located in the tumor stroma (*Figure 5G*, *Figure 5—figure supplement 2*). Furthermore, gene set variation analysis (GSVA) of upregulated genes revealed that TREM2_TAMs were enriched for PD1 signaling, innate immune system, and GC chemosensitivity, whereas Angio-TAMs were enriched for tumor angiogenesis, chemokines, HIF, VEGF pathway, and ECM organization involved in promoting the development of vascular and hypoxic tumors (*Figure 5H*). Furthermore, we found that TREM2_TAMs and Angio-TAM abundance were both highly correlated with Treg abundance (*Figure 5I*, *Figure 5—figure supplement 1B*), which supports the notion that TAMs play crucial roles in the modulation of Treg maturation and recruitment. Interestingly, the intercellular crosstalk analysis revealed different ligand–receptor axes between TREM2_TAM-Tregs and Angio-TAM-Tregs. The SPP1-CD44 axis and the 'do not eat me' axis THBS1-CD47, which is an immune checkpoint controlling CD8+ T cell activation and immune escape, were mainly enriched in Angio-TAM and Tregs. Additionally, we found Angio-TAM derived from HpGC or exHpGC showed higher expression of *THBS1*, *VEGFA*, and *SPP1* than Angio-TAM of non-HpGC, and HC (*Figure 5— figure supplement 1D*). The immune checkpoint axes NECTIN2-TIGIT, CD86-CD28, CD86-CTLA4, CXCL16-CXCR6, and LGALS9-HAVCR2 (TIM3) were exclusively enriched in TREM2_TAM and Tregs (*Figure 5J*, *Figure 5—figure supplement 1C*). Furthermore, the deconvolution analysis in TCGA bulk GC transcriptome dataset revealed that the enrichment of Angio-TAMs was more significant in HpGC compared with non-HpGC (p=0.047, Wilcoxon test; *Figure 5K*) and was associated with poor prognosis of GC (p=0.0035, log-rank test; *Figure 5L*). Overall, our results revealed the heterogeneity and function differences of myeloid cells of GC with different *H. pylori* infection status, and decipher distinct roles of TREM2_TAM and Angio-TAM involved in gastric carcinogenesis and prognosis.

## Transcriptomic diversity of cancer-associated fibroblasts in *H. pylori* infection-associated GC

Cancer-associated fibroblasts (CAFs), another major component of the TME, not only produce ECM but also secrete various molecules, including cytokines, chemokines, and growth factors, which are correlated with migration, infiltration, and EMT, thereby playing a role in GC development and resistance to immunotherapy (*Chen and Song, 2019*). Therefore, we focused on the molecular characteristics of CAFs. CAFs can be divided into six subpopulations, including three subsets of vascular-related CAF (vCAF) featuring the expression of microvasculature genes MCAM, RGS5, and MYH11; and two subsets of matrix CAF (mCAF) characterized by high expression of extracellular matrix (ECM) genes including *PDGFRA*, *LUM*, and *DCN* and inflammatory CAF (iCAF), which was identified by high expression of *TWIST1*, *VEGFA* and inflammatory chemokine genes *CXCL5/8*, *IL11* (*Nishina et al., 2021*), and *IL24* (*Figure 6A–C*, *Figure 6—figure supplement 1A* and *Supplementary file 6*). Interestingly, iCAF shared highly consistent transcriptome phenotype of iCAFs derived from human intrahepatic cholangiocarcinoma (*Figure 6—figure supplement 1B*; *Zhang et al., 2020*). The organization of the

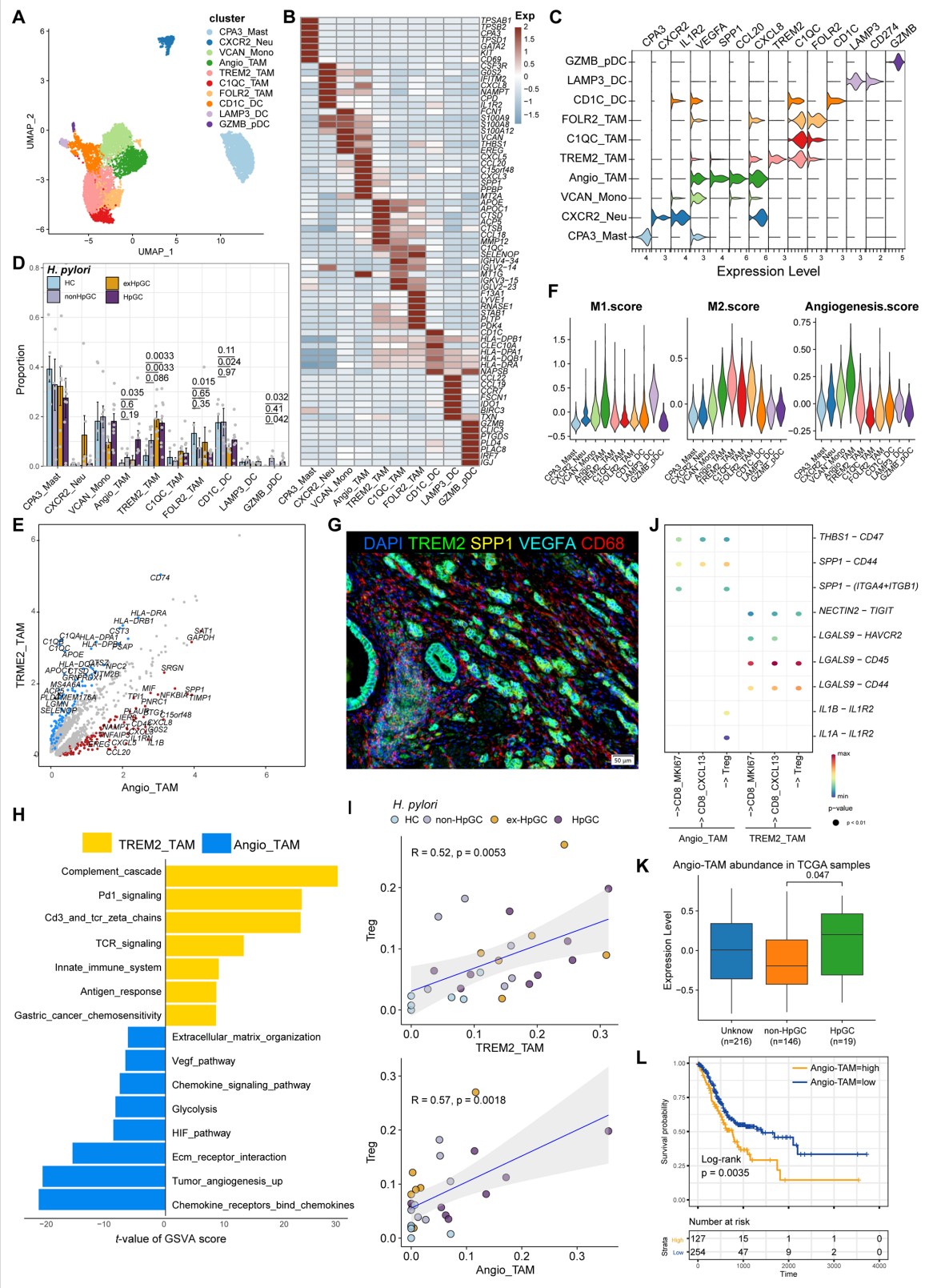

**Figure 5.** Characterization of tumor-infiltrating myeloid cells by scRNA-seq in *H. pylori* infection-associated gastric cancer (GC). (**A–C**) Unbiased clustering of myeloid cells generated nine clusters (**A**), and molecular features were annotated according to the top seven differentially expressed genes (DEGs) (**B**) and representative genes (**C**). (**D**) The percentage contribution of each myeloid cell cluster in HC, non-HpGC, ex-HpGC, and HpGC. p-Values were assessed by Student's *t*-test. (**E**) Volcano plot showing the DEGs between Angio-TAM and TREM2+ TAM. (**F**) Violin plot showing the expression of

*Figure 5 continued on next page*

*Figure 5 continued*

functional gene sets in myeloid clusters. (**G**) Immunostaining showing the distribution of Angio-TAM and TREM2⁺ TAM in one HpGC sample. (**H**) Bar plot showing the enriched signaling pathway between Angio-TAM and TREM2⁺ TAM. (**I**) The correlation of cell type (percentage) between Tregs and Angio-TAM and TREM2⁺ TAM. (**J**) Dotplot showing intercellular interactions among suppressive T cells and Angio-TAM and TREM2⁺ TAM. (**K**) The relative abundance of Angio-TAM in HpGC and non-HpGC in the The Cancer Genome Atlas (TCGA) stomach adenocarcinoma (STAD) dataset, p-values were assessed by the Wilcoxon test. (**L**) Kaplan–Meier survival analysis of TCGA STAD patients stratified by Angio-TAM relative abundance, which was used to group samples into high and low groups based on the 33rd and 67th percentiles. The p-value of the two-sided log-rank test is shown.

The online version of this article includes the following figure supplement(s) for figure 5:

**Figure supplement 1.** Myeloid cell characteristics in gastric cancer (GC) with different *H. pylori* infection status.

**Figure supplement 2.** Immunostaining showing the expression of Angio-TAM and TREM2⁺ TAM, respectively, in HpGC, ex-HpHC, non-HpGC, and HC.

CAF compartment showed great differences in GC with different *H. pylori* infection status (***Figure 6D***) and the ratio of iCAF was significantly higher in HpGC tissues than in normal gastric tissues (p=0.023, Student's *t*-test; ***Figure 6E***) and the ex-HpGC also showed an increase trend (p=0.083, Student's *t*-test; ***Figure 6E***). Further enrichment analysis revealed that iCAF enriched chemokines, TGFβ, ECM, IL6 pathway, PDGFRA pathway, inflammatory response, hypoxia, tumor angiogenesis, and VEGF signaling pathway (***Figure 6F***), which are crucial for tumor progression, metastasis, and immune escape (***Nishina et al., 2021***). Additionally, the abundance of iCAF significantly correlated with Angio-TAM and suppressive Treg cells (***Figure 6G***), and the above results indicated an important role of iCAF in tumor TME regulation and immune escape.

Further intercellular crosstalk analysis showed that iCAF had the strongest interaction with suppressive T cells and TAM, indicative of complex roles in the TME (***Figure 6H***). Specifically, the THBS, NECTIN, TIGIT, ANGPTL, and VEGF signaling pathways were enriched in the cellular interplay between iCAF and suppressive T cells and TAM (***Figure 6I***). Detailed ligand–receptor crosstalk analysis showed that iCAF interacts with CD8_CXCL13, Tregs mainly by NECTIN2-TIGIT ligand–receptor pair (***Figure 6I***), which strongly implied that *H. pylori* infection upregulates the expression of NECTIN2 in stromal cells that competitively bind to the TIGIT receptor on T cells, leading to the inhibition of T cell responses and immune escape of GC. In addition, angiogenesis-associated ligand–receptor pairs such as VEGFA/B-VEGFR1 and ANGPTL4-SDC2 were mainly enriched between iCAF and Angio-TAMs (***Figure 6I***). Interestingly, we found iCAF derived from HpGC showed higher expression of *NECTIN2*, *VEGFA*, *PVR* and inflammatory chemokine genes *IL11* and *IL24* compared to ex-HpGC, non-HpGC, and HC. Additionally, we found that the elevated expression level of *IL11*, *VEGFA*, *IL24*, and *TWIST1* was correlated with poor prognosis in GC (p=0.14, p<0.05, p<0.05, and p<0.05, respectively, log-rank test; ***Figure 6—figure supplement 1C and D***). Furthermore, the validation results by virtue of two public GC bulk RNA datasets (***Cristescu et al., 2015***; ***Bass and Thorsson, 2014***) revealed that the abundance of iCAF was elevated in HpGC samples than non-HpGC (p=0.035, Wilcoxon test; ***Figure 6J***) and was highly associated with poor prognosis in two public bulk transcriptomic dataset (p=0.12 and=0.007, respectively, log-rank test; ***Figure 6K***). The above results indicated that iCAFs promoted tumor angiogenesis and immune suppression in *H. pylori* infection-associated GC, by upregulation of VEGFA/B-VEGFR1 pathway, and NECTIN2-TIGIT pathway.

## Association of immune and stromal composition abundance with GC immunotherapy

*H. pylori* infection has multiple immunomodulatory effects on the host, which can not only activate the immune response but also negatively regulate it, causing immune escape. However, the key cell players and molecular features for predicting the GC immunotherapy response remains largely unknown. To reveal the detailed molecular features of the *H. pylori* infection-associated GC TME for predicting the outcomes of immunotherapy, we performed the devolution analysis to evaluate the cell type abundance, immune checkpoint, and angiogenic signature expression in GC immunotherapy-treated bulk RNA sequencing dataset (***Kim et al., 2018***). GSEA and devolution analysis demonstrated a high abundance of iCAF and Angio-TAM in anti-PD-1 immunotherapy non-responsive (NR) patients, while CD8_CXCL13, CD8_GZMB, CD8_IFNG, and CD8_MKI67 were more abundant in anti-PD-1-responsive cases (***Figure 7A*** and p<0.05, Student's *t*-test; ***Figure 7—figure supplement 1A***). The relative abundance of several cell clusters such as CD8_CXCL13, CD8_MKI67 had a remarkable

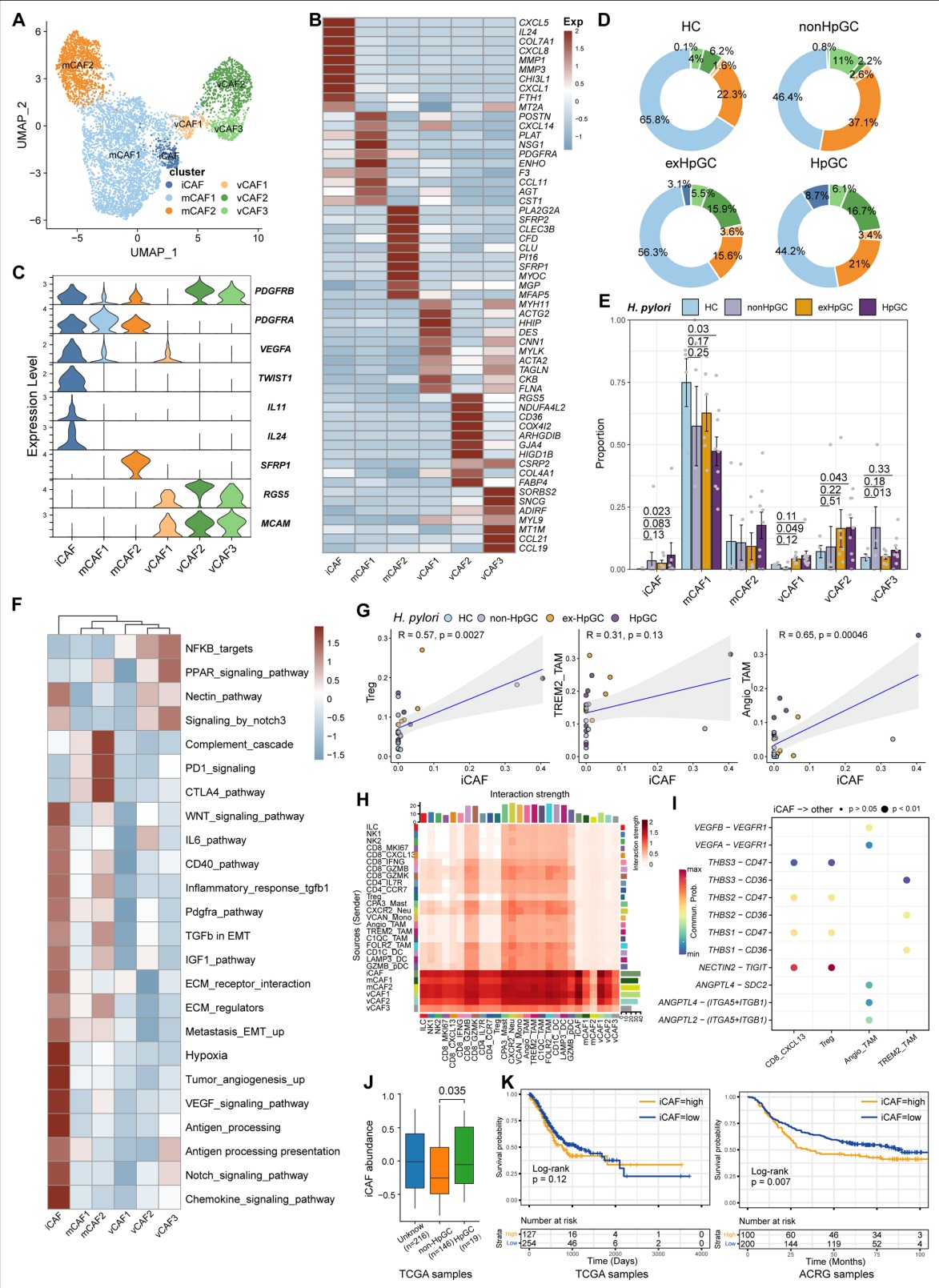

**Figure 6.** Characterization of cancer-associated fibroblasts (CAFs) by scRNA-seq in *H. pylori* infection-associated gastric cancer (GC). (**A**) Unbiased clustering of CAF generated six clusters. (**B, C**) Molecular features annotations according to the top ten differentially expressed genes (DEGs) (**B**) and representative genes (**C**). (**D**) The pie plot showing the abundance distribution of six cancer-associated fibroblast (CAF) subset in HC, non-HpGC, ex-HpGC, and HpGC. (**E**) The percentage contribution of each CAF cluster in HC, non-HpGC, ex-HpGC, and HpGC. p-Values were assessed by Student's

*Figure 6 continued on next page*

*Figure 6 continued*

t-test. (**F**) Heatmap showing the enriched signaling pathway among six CAF clusters. (**G**) The cell type percentage correlation between iCAF and Treg, Angio-TAM, and TREM2⁺ TAM. (**H**) Heatmap showing intercellular interaction strength among immune cells and different subsets of CAF. (**I**) Enriched signaling pathway among suppressive T cell, Angio-TAM, and TREM2⁺ TAM with iCAF. (**J**) The relative abundance of iCAF in HpGC and non-HpGC in TCGA STAD dataset. p-Values were assessed by Wilcoxon test. (**K**) Kaplan–Meier plot shows that the abundance of iCAF predicts poor prognosis of GC using two public bulk RNA sequencing dataset (left, TCGA; right, ACRG). The abundance of iCAF was used to group samples into high and low groups based on 33rd and 67th percentile. The p-value of two-sided log-rank test is shown. ACRG: Asian Cancer Research Group.

The online version of this article includes the following figure supplement(s) for figure 6:

**Figure supplement 1.** Cancer-associated fibroblast (CAF) subtypes characteristic in gastric cancer (GC) with different *H. pylori* infection status.

correlation with improved anti-PD-1 overall survival (OS) and progression-free survival (PFS), while the abundance of iCAF and Angio-TAM was significantly associated with poor survival (***Figure 7B and C***). Furthermore, we defined an anti-PD-1 immune signature and an angiogenic signature (***Figure 7D***, ***Figure 7—figure supplement 1B***) derived from the cell type signatures of scRNA-seq result and found that the immune signature was highly correlated with CD8_GZMK and CD8_GZMB, while the angiogenic signature highly correlated with iCAF and Angio-TAM (***Figure 7—figure supplement 1C***). Interestingly, we found that the expression of immune signature was higher in immunotherapy responsive GC cases than non-responsive cases, while the angiogenic signature showed the opposite trend (p<0.05, Wilcoxon test; ***Figure 7E***). To better understand the cellular and molecular characteristics that react to immunotherapy, we constructed a comprehensive model comprising cellular composition and signature to evaluate the immunotherapy response of different parameters (***Figure 7—figure supplement 1D and F***). The results showed that immune signatures such as *CXCL13*, *LAG3*, *TIGIT*, and *PDCD1*, and cell types such as CD8_CXCL13 and TREM2_TAM had high diagnostic power to distinguish between anti-PD-1-responsive cases and non-responsive cases (area under the curve [AUC] >0.75). Additionally, the immune signature we defined also effectively distinguished anti-PD-1-responsive cases from non-responsive cases to achieve better anti-PD-1 immunotherapy prognosis (AUC = 0.755, ***Figure 7F***), whereas the angiogenic signature had poor diagnostic ability for identifying anti-PD-1 responses as well as poor anti-PD-1 immunotherapy prognosis (AUC = 0.367, ***Figure 7F***). Further Kaplan–Meier survival analysis revealed that the defined immune signature was correlated with better immunotherapy efficacy in terms of OS and PFS in GC while the angiogenic signature linked with poor immunotherapy response (p<0.05, log-rank test; ***Figure 7G***, ***Figure 7—figure supplement 1***). In brief, by integrating our single-cell transcriptional results of GC with public bulk RNA-seq data of immunotherapy-treated GC, we identified several cell types and molecules that could serve as indicators to predict the immunotherapy response of GC, thus flagging up individualized GC therapy.

## Discussion

GC is the third most common tumor in Asia and is mainly caused by *H. pylori* infection. Numerous studies had shown that *H. pylori* infection not only led to GC but also affected the TME of GC and response to immunotherapy (***Oster et al., 2022***; ***Shi et al., 2022***; ***Hong et al., 2015***; ***Wroblewski et al., 2010***). Owing to advances in scRNA-seq, the heterogeneity of cell components in the TME of human malignancies can be elucidated. In this study, we profiled the single-cell transcriptional characteristics of the TME of GC tissues to reveal the role of *H. pylori* infection in GC development, TME regulation, and immunotherapy response. Specifically, we revealed the TME transcriptomic phenotype and composition, functional features, and potential intercellular interactions of GC with different *H. pylori* infection status, which provided evidence and biomarkers for predicting the prognosis of immunotherapies for *H. pylori* infection-associated GC.

First, we used inferCNV algorithms to distinguish non-malignant epithelium from the malignant epithelium and found that malignant epithelium derived from *H. pylori*-associated GC exhibited inflammation and EMT signatures, which provided potential biomarkers for the diagnosis of *H. pylori* infection-associated GC (***Chang et al., 2015***). We also revealed that *H. pylori* infection leads to dramatic loss or death of parietal cells and delineated the development trajectory of gastric chief cells along the cascade GC at the single-cell level, which broadened our previous findings to reveal the chief cell transitions (***Zhang et al., 2021***). Based on the expression pattern of signature genes and TFs, we found that chief cells could give rise to neck cells; subsequently, SPEM then showed distinct

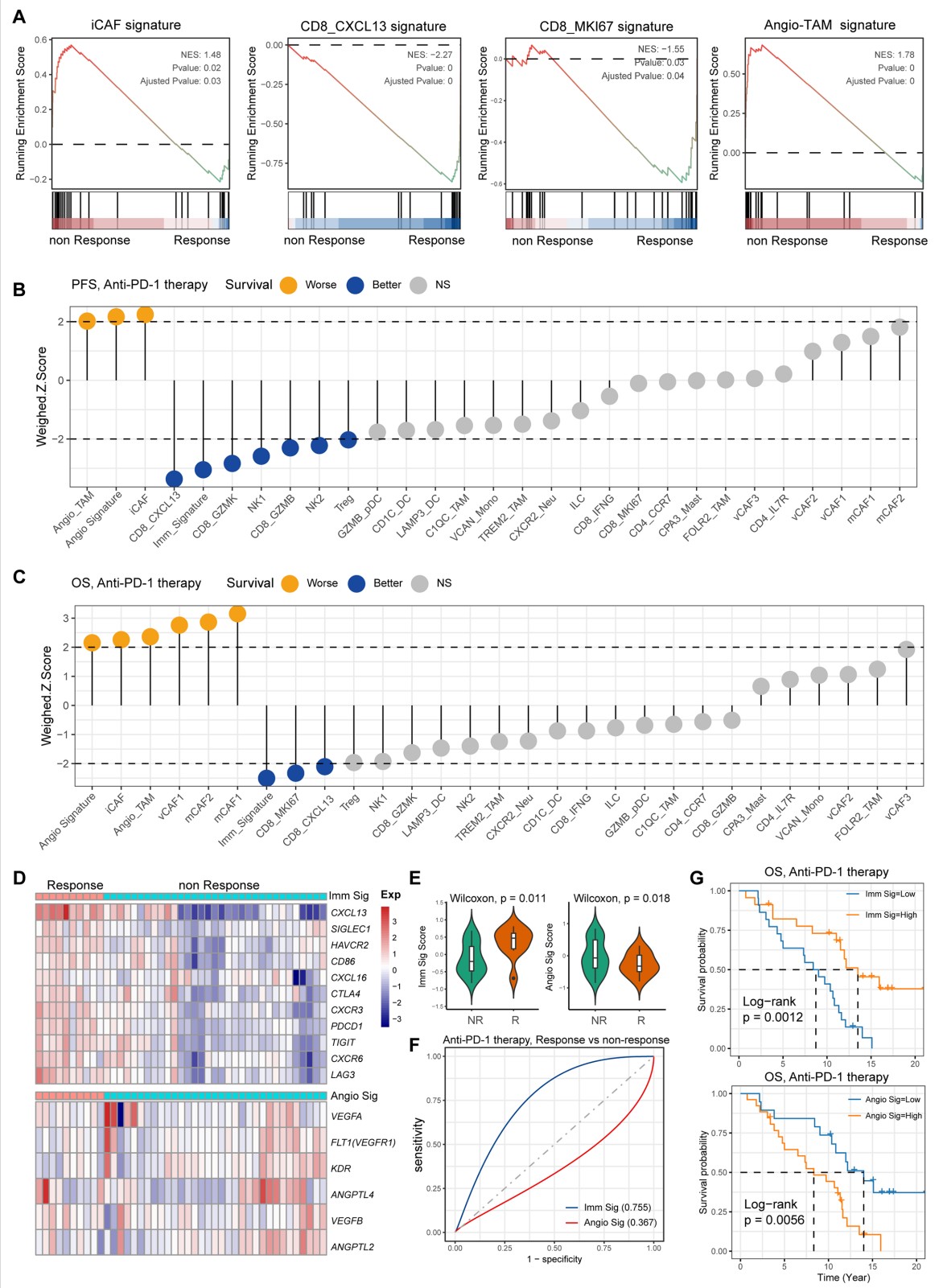

**Figure 7.** Single-cell tumor microenvironment (TME) composition associated with gastric cancer (GC) immunotherapy outcome. (**A**) Gene set enrichment analysis (GSEA) plot showing the enrichment of iCAF, CD8_MKI67, Angio-TAM, and CD8_CXCL13 in anti-PD-1-responsive or non-responsive GC. NES, normalized enrichment score. (**B, C**) Bar chart showing the cell subtypes relative abundance, the immune signature, angiogenesis signature derived from scRNA-seq predicted GC immunotherapy outcome, progression-free survival (**B**), and overall survival (**C**). (**D**) Heatmap showing the

*Figure 7 continued on next page*

*Figure 7 continued*

expression of immune and angiogenesis signature derived from scRNA-seq in immunotherapy responsive and non-responsive GC. (**E**) Violin plot showing the expression of immune signature and angiogenesis signature in immunotherapy responsive and non-responsive GC. The p-value of Wilcoxon test is shown. (**F**) A model for evaluating the GC immunotherapy sensitivity and specificity using the immune signature and angiogenesis signature derived from scRNA-seq. (**G**) The Kaplan–Meier plot showing the immune signature and angiogenesis signature could efficiently predict prognosis of GC anti-PD-1 therapy. The p-value of two-sided log-rank test is shown.

The online version of this article includes the following figure supplement(s) for figure 7:

**Figure supplement 1.** Gastric cancer (GC) tumor microenvironment (TME) characteristics related to GC immunotherapy response.

branches, and only part of SPEM developed into intestinal-specific cell types, including IM and enterocytes, under conditions of chronic inflammation, which further validates the evolutionary path of the non-malignant epithelium of GC under *H. pylori* infection.

Next, we evaluated the heterogeneity and molecular features of immune and stromal components among the TME of normal and all GC tissues and found that HpGC tissues were enriched with iCAF, Angio-TAM, TREM2[+] TAM, and suppressive T cells, and that iCAF was highly linked to Angio-TAM and suppressive T cells; also, the abundance of iCAF is associated with poor immunotherapy efficacy. Intercellular crosstalk analysis revealed that iCAF and TREM2[+]TAM interacted with suppressive T cells: Treg and CXCL13[+]CD8 via the NECTIN2-TIGIT pathway. Previous studies had shown that *H. pylori* infection upregulates PD-L1 expression in a variety of cells, including TAM, eosinophils, and dendritic cells (**Shi et al., 2022**; **Peng et al., 2020**; **Silva et al., 2016**). PD-L1 upregulation in these cells may compete with PD-1 binding to T cells, thereby inhibiting T cell responses and affecting the therapeutic efficacy of PD-1/PD-L1 inhibitors (**Holokai et al., 2019**; **Shen et al., 2019**). The above results indicated that *H. pylori* infection could upregulate NECTIN2 expression in immune and stromal components, which mediated immune escape and impaired immunotherapy efficacy. Hence, inhibition of the TIGIT-NECTIN2 axis may be a promising treatment strategy in HpGC.

*H. pylori* infection impairs the efficacy of anti-PD-1/CTLA4 immunotherapies or anticancer vaccines, and a lack of efficient targets impedes the treatment of *H. pylori* infection-associated GC (**Oster et al., 2022**). However, the key cellular players and signaling pathways involved in the association of *H. pylori* infection with immunotherapy efficacy remain poorly investigated. In this study, iCAFs exhibited upregulation of angiogenic molecules VEGFA/B and ANGPTL, with the highest expression levels identified in cells derived from HpGC. Additionally, iCAFs were notably enriched in tumor angiogenesis, ECM, TGFβ, and VEGF pathway, our results also showed that iCAFs could interact not only with immunosuppressive T cells through the NECTIN2-TIGIT and THBS1/2-CD47 axes, but also with Angio-TAMs through the VEGFA/B-VEGFR1 and ANGPTL-SDC2 axes. These results indicated an important role of iCAFs in stromal TME regulation, immune escape, and tumor angiogenesis. Recent research displayed that block of placental growth factor (PIGF), a member of the vascular endothelial growth factor, could inactive CAFs and reduce the deposition of fibrosis-associated collagen and desmoplasia in pancreatic cancer TME, which could improve survival rate of murine pancreatic ductal adenocarcinoma (PDAC) model. Furthermore, PD-L1-directed PIGF/VEGF block by fusing of single chain of anti-PD-L1 antibody (atezolizumab) to VEGF-Grab could target PD-L1-expressing CAF, thus promoting anti-fibrotic and antitumor effect (**Kim et al., 2022b**). As we know, CAFs can shape the ECM and form a barrier for drug or therapeutic immune cell penetration, which prevents the deep penetration of drugs and immune cells into tumor tissues, thereby reducing the effectiveness of treatments. Therefore, inhibition of tumors by modulating CAFs or overcoming their barrier effects is a new therapeutic strategy. Taken together, our findings were consistent with those of previous ones that had hypothesized that anti-angiogenic targeted therapy could augment antitumoral immunity and enhance immunotherapy efficacy by targeting tumor angiogenesis, exhausted T cells proliferation and myeloid cell inflammation (**Zhu et al., 2022**; **Fukumura et al., 2018**; **Hegde et al., 2018**; **Ragusa et al., 2020**).

This study has some limitations, and the main findings should be validated in a large patient cohort. Secondly, our findings were mainly based on scRNA-seq and bioinformatic analysis, and only a small number of key molecules and cell types have been verified by immunostaining; additionally, we conducted deconvolution analysis using external datasets to further validate our results. Even these validations provide support at genomic and protein level, and it is important to note that functional validation is still lacking. Further clinical trial research should be performed to validate the efficacy of

combination immunotherapy alone or in combination with anti-VEGF therapy in *H. pylori* infection-associated GC. Nevertheless, we provided vital insights into the molecular and cellular heterogeneity of *H. pylori*-associated GC at a single-cell level. Moreover, the enrichment of the TIGIT-NECTIN immune checkpoint and angiogenesis signaling pathways in different cell types in the TME of *H. pylori* infection-associated GC implies that the potential combination of immunotherapies and anti-angiogenic targeted therapeutic modalities could be an effective therapeutic approach for *H. pylori* infection-associated GC.

## Materials and methods

### Acquisition of fresh tissue materials and preparation of a single-cell suspension

Normal gastric tissue biopsy specimens were obtained from 2 to 3 gastric antrum sites through conventional upper gastrointestinal endoscopy using biopsy forceps. GC tissues were obtained immediately after surgical resection with a scalpel. The fresh tissue samples were then washed with phosphate-buffered saline (PBS) and divided into two parts. One part was processed into a single-cell suspension, and the second part was used for other experiments, including the detection of *H. pylori* DNA, H&E, and immunofluorescence staining. The patients' serum was used to detect the IgG antibody of *H. pylori*. Patients with positive serum antibody and genomic DNA of *H. pylori*, and *H. pylori* observed on H&E slides, were defined as GC patients with current *H. pylori* infection (HpGC). Patients with positive serum antibody but negative genomic DNA had a history of *H. pylori* eradication or antibiotic use before surgery, and *H. pylori* cannot be observed on H&E slides. They were defined as GC patients with previous *H. pylori* infection (ex-HpGC). The negative detection of serum antibody and genomic DNA, as well as H&E, indicated that the volunteers never had *H. pylori* infection, which was defined as GC patients or healthy volunteers without *H. pylori* infection (non-HpGC and HC).

The biopsy and surgical samples used to prepare the single-cell suspension were immediately put into a tissue preservation solution (Miltenyi Biotec, Germany) and transported to the field laboratory in an ice bath for immediate preparation of the single-cell suspension. Fresh tissue samples were washed with 4°C precooled Dulbecco's Phosphate-Buffered Salines (DPBS, Solarbio, Beijing) for 2–3 times, cut into small pieces with surgical scissors, and then transferred to a 1.5 ml centrifuge tube. The tissue samples were incubated in a shaker at 37°C for 30–50 min with an in-house prepared enzymolysis solution (1 mg/ml type IV collagenase [Solarbio] + 10 U/µl DNase I DNase I [Roches]) or MACS Human Tumor Dissociation kit (DS_130-095-929, Miltenyi Biotec). The incubation was terminated when the digestive fluid turned turbid and the tissue block disappeared. The cell suspension was filtered with a 40-µm cell sieve and centrifuged at 4°C at 300 r/min for 5 min. After the supernatant was discarded, the cells were resuspended in 1 ml of DPBS, and 3 ml of precooled red blood cell lysate (Solarbio) were added. The cells were evenly aspirated, incubated at 4°C for 5–10 min, and centrifuged again. The cells were stained with 7-aminoactinomycin D (7-AAD, eBioscience, Cat# 00-6993-50) staining solution (100 µl 1% BSA/PBS + 5 µl 7-AAD) at 25°C for 5 min. Consequently, the individual cells with high quality were sorted by FACS and performed with a BD Aira II instrument and checked by staining them with trypan blue under the microscope for single-cell transcriptomic library construction.

### Detection of *H. pylori* nucleic acid in the gastric mucosa

*H. pylori* in the human gastric mucosa was detected using a *H. pylori* nucleic acid detection kit (PCR-fluorescent probe method, Daan Gene, Guangzhou, China), according to the manufacturer's protocols. Briefly, the following steps were performed: (1) DNA extraction: DNA extraction solution was added to the negative/positive quality controls and the samples to be tested. The solutions were treated at a constant temperature of 100°C for 10 min and centrifuged at 12,000 r/min for 5 min for later use; (2) fluorescence PCR: 2 µl samples were briefly centrifuged at 8000 r/min and then placed in a PCR machine, 50°C for 8 min, 93°C for 2 min, 93°C for 45 s→55°C for 1 min (10 cycles), 93°C for 30 s→55°C for 45 s (30 cycles), and 50°C for 45 s to collect fluorescence values. The system automatically analyzed the results to calculate the threshold cycle value (CT value); (3) quality control and results calculation: the fluorescence signal of the negative control material did not increase, the growth curve of the positive control material was S-shaped, and the Ct value was ≤ 27.04. The sample results were compared to those of the quality control material. *H. pylori* cagA was detected with the Nucleic acid

test kit of *H. pylori* type I (fluorescent PCR method, Beijing Xinji Yongkang, China), and the method was carried out according to the manufacturer's protocols.

## Detection of serum antibody against *H. pylori*

*H. pylori* IgG ELISA kit (IBL, Germany) was used to detect the presence of anti-*H. pylori* antibodies in the serum of the subjects, following the manufacturer's instructions. The cut-off index (COI) was determined by dividing the OD at 450 nm value of the sample by the $OD_{450}$ value of the cut-off standard. The serum was considered positive for *H. pylori* if the COI was > 1.2, while the serum was considered negative if the COI < 0.8.

## Single-cell sequencing and pre-processing data

We prepared single-cell RNA-seq libraries on the Chromium platform (10X Genomics, Pleasanton, CA) using the Chromium Next GEM Single-Cell 3' Kit v2 following the manufacturer's protocol to generate a complementary deoxyribonucleic acid (cDNA) library in Biomarker Technologies and Capitalbio Technology Corporation (China). Briefly, viable cells (7-AAD negative) with high quality were pooled together and washed thrice with RPMI-1640, concentrated to 700–1000 cells per μl, and then immediately loaded onto a 10×microfluidic chip (10X Genomics, v4) to generate single-cell mRNA libraries and then sequenced across six lanes on an Illumina X Ten or NovaSeq 6000 system (Illumina, Inc, San Diego, CA). Raw sequencing data were aligned to the GRH38 reference genome using the cellranger (10X Genomics, v4) count function. The count matrixes of gene expression from each sample were imported into the Seurat v4.1 (*Stuart et al., 2019*). We selected high-quality cells for further analysis following three measurements: (1) cells had either over 2001 unique molecular identifiers (UMIs), fewer than 6000 or more than 301 expressed genes or fewer than 20% UMIs derived from the mitochondrial genome; (2) genes expressed in more than 10 cells in a sample; and (3) cell doublets were removed using the DoubletFinder R package (v2.0.3) (*McGinnis et al., 2019*). The cell-by-gene expression matrixes of the remaining high-quality cells were integrated with the RunFastMNN function provided by the SeuratWrappers R package (v0.4.0) and then normalized to the total cellular UMI count. The union of the top 2000 genes with the highest dispersion for each dataset was used to generate an integrated matrix. We then performed data normalization, dimension reduction, and cluster detection as previously reported (*Zhang et al., 2021*). Briefly, the gene expression matrices were scaled by regressing the total cellular UMI counts and percentage of mitochondrial genes. Principal component analysis was conducted using highly variable genes, and the top 30 significant principal components were selected to perform Uniform Manifold Approximation and Projection (UMAP) dimension reduction, and visualization of gene expression. We annotated cell subclusters with similar gene expression patterns as the same cell type, and cell types in the resulting two-dimensional representation were annotated to known biological cell types using canonical marker genes.

## Detection of single-cell CNVs

We employed the inferCNV (*Tirosh et al., 2016*) R package (https://github.com/broadinstitute/inferCNV/wiki, v1.22.0, *Georgescu C and Haas, 2025*) to distinguish malignant and non-malignant epithelia of GC, and the initial CNV signal of each region was estimated based on the expression level from the scRNA-seq results with default parameters.

## Pathway enrichment and cell type abundance deconvolution analysis

To illustrate the enriched signaling pathways of fibroblast and myeloid subtypes, we used the GSVA (v2.0.6) (*Hänzelmann et al., 2013*) package to assess pathway differences using the C2 curated gene set provided by the Molecular Signatures Database, which was calculated with a linear model offered by the limma package. We used the GSVA (*Hänzelmann et al., 2013*) package to calculate the abundance of malignant epithelium subclusters in *H. pylori*-positive GC using the TCGA (*Bass and Thorsson, 2014*) and Asian Cancer Research Group (ACRG) cohorts (*Cristescu et al., 2015*). We used GSEA package to calculate the distribution of gene sets and cell type-specific signatures (top 30 DEGs) in lists of genes ordered by population expression differences.

## Malignant epithelium differentiation score

To assess the malignant epithelium differentiation heterogeneity at single cell level, we included a tumor differentiation-associated signature to evaluate the GC differentiation heterogeneity based on our previous study (*Zhang et al., 2021*), including PHGR1, MUC13, MDK, KRT20, LGALS4, GPA33, CLDN3, CLDN4, and CDH17, and then we employed the 'AddModuleScore' function of Seurat R package to calculate the differentiation degree of each tumor cell based on the expression level of differentiation-associated signature.

## TCGA data analysis

For the analysis of the correlation of tumor differentiation score and cell subtypes abundance with patients' clinical outcome, we employed the GSVA to calculate the differentiation score and cell subtypes abundance in each TCGA stomach adenocarcinoma (STAD) sample based on the tumor differentiation signature and the top 30 DEGs of each cell subtype, and then grouped samples into high and low groups based on 67th and 33rd percentile, respectively (*Cheng et al., 2021*). Kaplan–Meier survival curves were plotted using the R package 'survminer' (v0.4.9).

## Immunotherapy response and prognosis analysis

GC immunotherapy bulk RNA-seq data, along with curated clinical data from patients, were obtained from a previous study (*Kim et al., 2018*). To evaluate the association between the signatures of immune and stromal subtypes identified in our study and immunotherapy response to GC patient survival, the GSVA (*Cheng et al., 2021*) was used to calculate the combined expression value of the cell type-specific signatures (top 30 DEGs). We classified the patients into high and low groups based on the 50th percentile of cell subtypes abundance. Kaplan–Meier survival curves were plotted using the R package 'survminer'. Subsequently, a Cox proportional hazards model was conducted that included age and tumor stage. The results of the Cox regression model among the cell subtypes were visualized using weighted Z-scores.

## Trajectory analysis

To explore the potential differentiation routines between the non-malignant epithelium of GC, we performed trajectory analysis using the monocle R package as previously reported (v3.0) (*Trapnell et al., 2014*). First, we constructed the monocle object using the 'newCellDataSet' function, and the differentially expressed genes calculated via the 'differentialGeneTest' function were selected for trajectory analysis. Then 'DDRTree' function was used for dimensionality reduction and the 'plot_cell_trajectory' function for visualization.

## Intercellular crosstalk

We used the Cellchat (*Jin et al., 2021*) package (v0.0.2) to infer the intercellular communications and significant ligand–receptor pair of the *H. pylori* infection-associated GC TME, following a standard pipeline implemented in R (https://github.com/sqjin/CellChat; *Jin, 2023*). We first set the ligand–receptor interaction list in humans and projected the gene expression data onto the protein–protein interaction network by identifying the overexpressed ligand–receptor interactions. To obtain biologically significant cell–cell communication, the probability values for each interaction were calculated by performing permutation tests. The inferred intercellular communication network of each ligand–receptor pair and each signaling pathway was summarized and visualized using circle plots and heatmaps.

## Multilabeled immunofluorescence staining and multispectral imaging

A PANO 7-plex IHC kit (cat# 10004100100; Panovue, Beijing, China) was used to perform multiplexed immunofluorescence staining. Slides were placed in a 65°C oven overnight to for deparaffinization, and tissues were sequentially treated with xylene, ethanol, and distilled water. Slides were then microwaved (with antigen retrieval solution [citric acid solution, pH6.0/pH9.0]) and sequentially incubated with different primary antibodies and horseradish peroxidase-conjugated secondary antibodies (100 µl/tissue, *Supplementary file 7*). This incubation was followed by tyramide signal amplification (TSA Fluorescence Kits, Panovue). The slides were washed with 1× PBST after each incubation and microwaved after each round of TSA. After all the antigens were labeled, the nuclei were stained

with 4'–6'-diamidino-2-phenylindole (DAPI, Sigma-Aldrich, MO). Stained slides were scanned using a Mantra system (PerkinElmer, Waltham, MA) to obtain multispectral images.

## Acknowledgements

We thank all patients enrolled in this study. All authors declare that they have no competing interests. This research was funded by National Science and Technology Major Project for Prevention and Treatment of Infectious Diseases (2018ZX10101003-005-005), and this study was also supported by grants from National Natural Science Foundation of China (82203636) and Young Talent Project of Chinese PLA General Hospital (20230403).

## Additional information

### Funding

| Funder | Grant reference number | Author |
| --- | --- | --- |
| National Science and Technology Major Project | Prevention and Treatment of Infectious Diseases (2018ZX10101003-005-005) | Chunjie Liu |
| National Natural Science Foundation of China | 82203636 | Min Zhang |
| Young Elite Scientist Sponsorship Program by CAST | 2022QNRC001 | Min Zhang |

The funders had no role in study design, data collection and interpretation, or the decision to submit the work for publication.

### Author contributions

Xin Zhang, Investigation, Methodology, Writing – original draft, Writing – review and editing; Guangyu Zhang, Data curation, Software, Methodology, Writing – original draft, Writing – review and editing; Shuli Sang, Resources, Formal analysis, Investigation, Methodology; Yang Fei, Xiaopeng Cao, Resources, Investigation, Methodology; Wenge Song, Feide Liu, Haoxia Tao, Hongwei Wang, Resources, Investigation; Jinze Che, Yiyan Guan, Shipeng Rong, Investigation; Lihua Zhang, Resources; Lijuan Pei, Investigation, Methodology; Sheng Yao, Conceptualization, Methodology, Writing – review and editing; Yanchun Wang, Conceptualization, Investigation, Methodology, Project administration, Writing – review and editing; Min Zhang, Data curation, Software, Validation, Visualization, Writing – review and editing; Chunjie Liu, Conceptualization, Funding acquisition, Methodology, Project administration, Writing – review and editing

### Author ORCIDs

Yanchun Wang http://orcid.org/0000-0002-0203-7312
Min Zhang https://orcid.org/0000-0002-7785-5475

### Ethics

The study was conducted according to the guidelines of the Declaration of Helsinki and approved by the Ethics Committee of Fourth Medical Center of PLA General Hospital. (2021KY011-HS001, 2021KY041-HS001) All study participants provided written informed consent.

Reviewer #1 (Public review): https://doi.org/10.7554/eLife.99337.3.sa1
Reviewer #2 (Public review): https://doi.org/10.7554/eLife.99337.3.sa2
Author response https://doi.org/10.7554/eLife.99337.3.sa3

# Additional files

## Supplementary files

Supplementary file 1. Top 30 DEGs of nine main cell types.

Supplementary file 2. Top 30 DEGs of six malignant epithelium subclusters.

Supplementary file 3. Top 30 DEGs of nine non-malignant epithelium subclusters.

Supplementary file 4. Top 30 DEGs of T/NK subclusters.

Supplementary file 5. Top 30 DEGs of myeloid cells subclusters.

Supplementary file 6. Top 30 DEGs of cancer-associated fibroblasts subclusters.

Supplementary file 7. Multilabel immunofluorescence staining antibody.

MDAR checklist

## Data availability

The processed scRNA-seq data required to reproduce the analysis and figures have been deposited on Zenodo, accessed with link https://zenodo.org/record/8082331. The TCGA GC bulk RNA-seq expression data were downloaded from the UCSC Xena website (https://xenabrowser.net/) and the GC immunotherapy expression matrix data can be accessed in the European Nucleotide Archive with accession PRJEB25780. Other GC bulk RNA and microarray data used in this study are available in the NCBI database under the accession code GEO accession numbers GSE62254 and GSE2669. All data needed to evaluate the conclusions in the paper are present in the paper and/or the Supplementary Materials.

The following previously published datasets were used:

| Author(s) | Year | Dataset title | Dataset URL | Database and Identifier |
|---|---|---|---|---|
| Zhang M | 2024 | Uncovering prognostic architecture of response to immunotherapy in H. pylori infection associated gastric cancer by single-cell transcriptomic analysis | https://doi.org/10.5281/zenodo.8082331 | Zenodo, 10.5281/zenodo.8082331 |
| Kim ST, Cristescu R, Bass AJ, Kim KM, Odegaard JI, Kim K, Liu XQ, Sher X, Jung H, Lee M, Lee S, Park JO, Park YS, Lim HY, Lee H, Choi M, Talasaz A, Kang PS, Cheng J, Loboda A, Lee J, Kang WK | 2018 | Pembrolizumab in metastatic gastric cancer: comprehensive molecular characterization of clinical response | https://www.ebi.ac.uk/ena/browser/view/PRJEB25780 | EBI European Nucleotide Archive, PRJEB25780 |
| Cristescu R, Lee J, Nebozhyn M, Kim K, Ting JC, Wong SS, Liu J, Yue YG, Wang J, Yu K, Ye XS, Do I, Liu S, Gong L, Fu J, Jin JG, Choi MG, Sohn TS, Lee JH, Bae JM, Kim ST, Park SH, Tan P, Ronghua C, Hardwick J, Kang WK, Ayers M, Hongyue D, Reinhard C, Aggarwal A, Kim S, Loboda A | 2015 | Molecular analysis of gastric cancer identifies discrete subtypes associated with distinct clinical characteristics and survival outcomes: the ACRG (Asian Cancer Research Group) study [gastric tumors] | https://www.ncbi.nlm.nih.gov/geo/query.cgi?acc=GSE62254 | NCBI Gene Expression Omnibus, GSE62254 |
| Boussioutas A, Bowtell D | 2005 | Expression profiling of Gastric Cancers | https://www.ncbi.nlm.nih.gov/geo/query/acc.cgi?acc=GSE2669 | NCBI Gene Expression Omnibus, GSE2669 |

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
