## [Editor Report · eLife Assessment]

This study presents a **valuable** description of the cellular and transcriptional landscape of the tumor microenvironment in 27 gastric cancer (GC) patients based on their *H. pylori* status (HpGC, ex-HpGC, non-HpGC). The single-cell RNA sequencing dataset and computational analysis are **convincing** and provide a starting point that is of value for understanding *H. pylori*-associated GC cell type composition, cell transitions, and mechanisms of response to therapy. The section correlating immunotherapy outcomes with GC cell type compositions from bulk RNAseq would have been strengthened by further comparing *H. pylori* GC versus non-*H. pylori* GC.

---

## [Referee Report · Reviewer #1 (Public review)]

In this study, the authors conducted a single-cell RNA sequencing analysis of the cellular and transcriptional landscape of the gastric cancer tumor microenvironment, stratifying patients according to their H. pylori status into currently infected, previously infected and non-infected patients. The authors comprehensively dissect various cellular compartments, including epithelial, stromal and immune cells and describe specific cell types and signatures to be associated with H. pylori infection, including (i) inflammatory and EMT signatures in malignant epithelial cells, (ii) inflammatory CAFs in stromal cells, (iii) Angio-TAMs, TREM2+ TAMs, exhausted and suppressive T cells in immune cells. Looking at ligand-receptor interactions as well as correlations between cell type abundances, they suggest that iCAFs interact with immunosuppressive T cells via a NECTIN2-TIGIT axis, as well as Angio-TAMs through a VEGFA/B-VEGFR1 axis and thereby promote immune escape, tumor angiogenesis and resistance to immunotherapy.

The authors conduct a comprehensive and thorough analysis of the complex tumor microenvironment of gastric cancer, both single-cell RNA sequencing data as well as the analysis seem of high quality and according to best practices. The authors validate their findings using external datasets and include some prognostic value of the identified signatures and cell types. Furthermore, they validate some of their findings using immunofluorescence. While the authors confirm key transcriptional signatures in external cohorts comparing HP infected and non-infected cases, the main conclusions drawn from their own patient cohort are based on the comparison between HPGC and healthy controls. This approach does not fully resolve which signatures and cell types are specifically driven by H. pylori infection. As the authors also acknowledge in the limitations of their studies, their conclusions would benefit from functional validation.

In summary, this study provides a valuable resource of the cellular and transcriptional heterogeneity of the tumor microenvironment in gastric cancers, distinguishing between positive, negative and previously positive HP infected gastric cancer patients. Given that HP is the main risk factor for gastric cancer development, the study provides valuable insights into potential HP driven transcriptional signatures and how these might contribute to this increased risk. However, the study would highly benefit from a clearer and more systematic comparison between HPGC and non-HPGC to better delineate infection-specific effects.

---

## [Referee Report · Reviewer #2 (Public review)]

Summary:

This study aims the describe the single-cell transcriptomes of H pylori-associated (Hp) gastric cancers and tumour microenvironment (TME), as a starting point to understand TME diversity stratified by Hp status.

RNAseq was performed for gastric cancers with current Hp+ (from N=9 people), ex-Hp+ (N=6), non-Hp (N=6), and healthy gastric tissue (N=6).

The study expands on previous single-cell transcriptomic studies of gastric cancers and was motivated by previous observations about the effect of H pylori status on therapeutic outcomes. The study includes a brief review of previous work and provides valuable context for this study.

Strengths:

The observations are supported by solid RNAseq study design and analysis. The authors describe correlations between Hp status and inferred molecular characteristics including cell lineages, enrichment for cell subclusters identifed as tumour-infiltrating lyphocyte cell types, tumour-infiltrating myeloid cells and cancer-associated fibroblasts.

The observed correlations between Hp status and enrichment of cell subclusters were broadly corroborated using comparisons to deconvolved bulk RNAseq from publicly available gastric cancer data, providing a convincing starting point for understanding the diversity of tumour microenvironment by Hp-status.

Weaknesses:

The authors acknowledge several limitations of this study.

The correlations with HP-status are based on a small number of participants per Hp category (N=9 with current Hp+; N=6 for ex-HP+ and non-HP), and would benefit from further validation to establish reproducibility in other cohorts.

The ligand-receptor cross-talk analysis and the suggestion that suppressive T cells could interact with the malignant epithelium through TIGIT-NECTIN2/PVR pairs, are preliminary findings based on transcriptomic analysis and immunostaining and will require further validation.

---

## [Author Response]

The following is the authors’ response to the original reviews

**Public Reviews:**

**Reviewer #1 (Public Review):**
In this study, the authors conducted a single-cell RNA sequencing analysis of the cellular and transcriptional landscape of the gastric cancer tumor microenvironment, stratifying patients according to their H. pylori status into currently infected, previously infected, and non-infected patients. The authors comprehensively dissect various cellular compartments, including epithelial, stromal, and immune cells, and describe specific cell types and signatures to be associated with H. pylori infection, including (i) inflammatory and EMT signatures in malignant epithelial cells, (ii) inflammatory CAFs in stromal cells, (iii) Angio-TAMs, TREM2+ TAMs, exhausted and suppressive T cells in immune cells. Looking at ligand-receptor interactions as well as correlations between cell type abundances, they suggest that iCAFs interact with immunosuppressive T cells via a NECTIN2-TIGIT axis, as well as Angio-TAMs through a VEGFA/B-VEGFR1 axis and thereby promote immune escape, tumor angiogenesis and resistance to immunotherapy.

We sincerely appreciate the Reviewer's interest in our study and their valuable insights on how we can further enhance our work.

The authors conduct a comprehensive and thorough analysis of the complex tumor microenvironment of gastric cancer, both single-cell RNA sequencing data as well as the analysis seem of high quality and according to best practices. The authors validate their findings using external datasets, and include some prognostic value of the identified signatures and cell types. However, most of their conclusions throughout the manuscript are based on the comparison between HPGC and healthy controls, which is not a valid comparison to determine which of the phenotypes are specifically driven by HP infection, e.g. Tregs are high in all GC types, independent of HP status. The same holds true for TREM+ TAMs and iCAFs, which are higher in GC in general. This makes it very difficult to assess the actual HP-driven signatures and cell types. Also, when looking at the correlation/transcriptional differences across different cell types and cellular interactions, the authors do not explicitly define if they are looking at the whole dataset (including healthy controls?) or only at certain patients (HPGC?), which again makes it difficult to interpret the results.

We sincerely appreciate the reviewer's thorough assessment and valuable feedback on our study. During our analysis, although we did not specifically identify cell types unique to non-HpGC, ex-HpGC, or HpGC, we found that TREM+ TAMs and iCAFs were enriched in *H. pylori*-infected GC, with an even higher proportion in HpGC. This suggests that the enrichment of TREM+ TAMs and iCAFs is correlated with *H. pylori* infection status.

However, gastric cancer is driven by multiple complex factors, including environmental influences, genetic mutations, and pathogenic infections. As single factor, the *H. pylori* infection does not significantly alter T cell proportions at the cellular level; rather, it affects the expression of immune checkpoint molecules (Author response image 1A-B). Importantly, we evaluated key molecules mediating the interaction among the iCAF with the angio-TAM and Tregs, the results show that the expression of NECTIN, PVR, VEGF, IL11 and IL24 are higher in ex-HpGC compared to the non-HpGC, with the highest expression observed in HpGC, which further validate the *H. pylori* -driven signatures (Author response image 1C).

The correlation analysis among different cell types was conducted within different groups based on their *H. pylori* infection status (Author response image 1C). However, transcriptional differences across different cell types and cellular interactions were analyzed using the entire dataset, including healthy controls. This approach ensured an unbiased identification of molecular and cellular-level differences among cell subtypes before determining whether these subtypes originated from HpGC or ex-HpGC.

**Author response image 1. sa3fig1:** A. The dot plot illustrates the enrichment of the TIGIT-PVR/NECTIN axis in the interaction between malignant epithelial cells and immunosuppressive T cells. B. T Dotplot showing the expression of NECTIN2 and PVR in non-HpGC, ex-HpGC, and HpGC cells. C. The bubble plot showing the expression of NECTIN, PVR, VEGF, IL11 and IL24 in the CAF within non-HpGC, ex-HpGC, and HpGC sample. D. The correlation of cell type (percentage) between Tregs, Angio-TAM, TREM2+ TAM and iCAF.

The authors aim to confirm some of their findings via immunofluorescence, which in principle is a great approach to validate their results. However, to be able to conclude that e.g. suppressive TIGIT+ T cells are located close to NECTIN2+ malignant epithelium and that this might facilitate immune escape in HPGC (Figure 4K), the authors should include stains that show that this is not the case in the other groups (nonHPGC, exHPGC and HC). The same holds true for Figure 5G.

Thank you for your valuable feedback. We have add the immunostaining of the ligand TIGIT and the receptor NECTIN2 on suppressive T cells and on the malignant epithelium, as well as signature marker of Angio-TAM and TREM2+ TAM including TREM2, SPP1, VEGF and CD68, in the non-HpGC, ex-HpGC and HC sample (Figure S3, Figure S5). We could find that TIGIT and NECTIN2 exclusively express in HpGC and ex-HpGC samples compared with non-HpGC and HC, with extremely higher in HpGC. Furthermore, the Angio-TAM and TREM2+ TAM were exclusively enriched in HpGC and ex-HpGC samples, barely expressed in non-HpGC and HC. The above results also support our finding that the H.p infection statue determinate the enrichment of Angio-TAM and TREM2+ TAM, also the interaction between suppressive T cells and malignant epithelium guided by TIGIT-NECTIN.

In summary, this study provides a valuable resource on the cellular and transcriptional heterogeneity of the tumor microenvironment in gastric cancers, distinguishing between positive, negative, and previously positive HP-infected gastric cancer patients. Given that HP is the main risk factor for gastric cancer development, the study provides valuable insights into HP-driven transcriptional signatures and how these might contribute to this increased risk, however, the study would highly benefit from a clearer and more stringent comparison between HPGC and nonHPGC.
**Reviewer #2 (Public Review):**
Summary:This study aims to describe the single-cell transcriptomes of H pylori-associated (Hp) gastric cancers and tumor microenvironment (TME), as a starting point to understand TME diversity stratified by Hp status. RNA-seq was performed for gastric cancers with current Hp+ (from N=9 people), ex-Hp+ (N=6), non-Hp (N=6), and healthy gastric tissue (N=6).The study expands on previous single-cell transcriptomic studies of gastric cancers and was motivated by previous observations about the effect of H pylori status on therapeutic outcomes. The study includes a brief review of previous work and provides valuable context for this study.

We thank the Reviewer for recognizing the interest of the topic, and for sharing their views on how we might further strengthen our work.

Strengths:The observations are supported by solid RNAseq study design and analysis. The authors describe correlations between Hp status and inferred molecular characteristics including cell lineages, enrichment for cell subclusters identified as tumour-infiltrating lymphocyte cell types, tumour-infiltrating myeloid cells, and cancer-associated fibroblasts.The observed correlations between Hp status and enrichment of cell subclusters were broadly corroborated using comparisons to deconvolved bulk RNAseq from publicly available gastric cancer data, providing a convincing starting point for understanding the diversity of tumour microenvironment by Hp-status.Weaknesses:The authors acknowledge several limitations of this study.The correlations with HP-status are based on a small number of participants per Hp category (N=9 with current Hp+; N=6 for ex-HP+ and non-HP), and would benefit from further validation to establish reproducibility in other cohorts.

Thank you for your valuable suggestions. We acknowledge that this may limit the generalizability and statistical power of our findings. However, despite the limited sample size, our analysis revealed statistically significant trends (e.g., p-value < 0.05) or consistent patterns in the data. The sample size in this study was constrained by the availability of participants meeting the inclusion criteria, particularly in the ex-HP+ and non-HP groups. We view these findings as hypothesis-generating and aim to validate them in future studies with larger cohorts.

The ligand-receptor cross-talk analysis and the suggestion that suppressive T cells could interact with the malignant epithelium through TIGIT-NECTIN2/PVR pairs, are preliminary findings based on transcriptomic analysis and immunostaining and will require further validation.

We appreciate the reviewer's comment and agree that the ligand-receptor cross-talk analysis and the proposed interaction between suppressive T cells and malignant epithelium via TIGIT-NECTIN2/PVR pairs are preliminary findings. These insights were derived from transcriptomic data and immunostaining, which provide valuable but indirect evidence of potential interactions. Our analysis revealed co-expression patterns of TIGIT in suppressive T cells and NECTIN2/PVR in malignant epithelial cells, and immunostaining demonstrated spatial proximity between these cell types. Previous studies have established the functional significance of TIGIT-NECTIN2/PVR interactions in immune regulation (PMID: 19815499, 27978489), supporting the biological plausibility of our observations. While our current data provide a foundation for this hypothesis, future studies involving functional assays or in vivo models would be valuable to confirm the biological relevance of these interactions. We view these findings as exploratory and aimed at guiding future research into the role of suppressive T cells in the tumor microenvironment.

**Recommendations for the authors:**

**Reviewer #2 (Recommendations For The Authors):**
(1) Software versions are missing from the scRNAseq section of the Methods.

Thank you for your feedback. The bioinformation analysis are performed by Seurat 4.1 version, we have annotated the software version in the revised manuscript.

(2) There is a data link to a deposit in Zenodo, subject to data access request to the authors. Do the authors intend to publish the scRNAseq data?

Thank you for your inquiry regarding the data availability. We fully intend to make the scRNA-seq data publicly accessible. Currently, the dataset has been deposited in Zenodo and is available upon request to ensure compliance with institutional and ethical guidelines. We are in the process of finalizing the necessary approvals for unrestricted public release. Once completed, we will update the Raw data with an open-access link to facilitate direct download.